# Graph Neural Networks as Gradient Flows

## Abstract

Dynamical systems minimizing an energy are ubiquitous in geometry and physics. We propose a gradient flow framework for GNNs where the equations follow the direction of steepest descent of a learnable energy. This approach allows to analyse the GNN evolution from a multi-particle perspective as learning attractive and repulsive forces in feature space via the positive and negative eigenvalues of a symmetric 'channel-mixing' matrix. We perform spectral analysis of the solutions and conclude that gradient flow graph convolutional models can induce a dynamics dominated by the graph high frequencies, which is desirable for heterophilic datasets. We also describe structural constraints on common GNN architectures allowing to interpret them as gradient flows. We perform thorough ablation studies corroborating our theoretical analysis and show competitive performance of simple and lightweight models on real-world homophilic and heterophilic datasets.

## 1 Introduction and motivations

Graph neural networks (GNNs) [38, 20, 21, 36, 7, 15, 27] and in particular their Message Passing formulation (MPNN) [19] have become the standard ML tool for dealing with different types of relations and interactions, ranging from social networks to particle physics and drug design. One of the often cited drawbacks of traditional GNN models is their poor 'explainability', making it hard to know why and how they make certain predictions [46, 47], and in which situations they may work and when they would fail. Limitations of GNNs that have attracted attention are over-smoothing [29, 30, 8], over-squashing and bottlenecks [1, 40], and performance on heterophilic data [31, 51, 13, 4, 45] – where adjacent nodes usually have different labels.

**Contributions.** We propose a *Gradient Flow Framework* (GRAFF) where the GNN equations follow the direction of steepest descent of a *learnable energy*. Thanks to this framework we can (i) interpret GNNs as a multi-particle dynamics where the learned parameters determine pairwise attractive and repulsive potentials in the feature space. This sheds light on how GNNs can adapt to heterophily and explains their performance and the smoothness of the prediction. (ii) GRAFF leads to residual convolutional models where the *channel-mixing* **W** is performed by a shared symmetric bilinear form inducing attraction and repulsion via its positive and negative eigenvalues, respectively. We theoretically investigate the interaction of the graph spectrum with the spectrum of the channel-mixing, proving that if there is more mass on the negative eigenvalues of **W**, then the dynamics is dominated by the graph-high frequencies, which could be desirable on heterophilic graphs. We also extend results of [29, 30, 8] by showing that when we drop the residual connection intrinsic to the gradient flow framework,

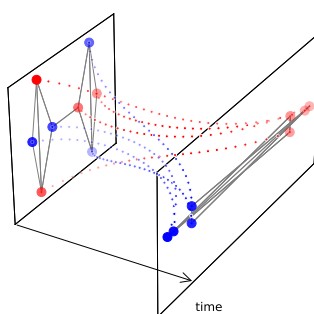

Figure 1: GRAFF dynamics: attractive and repulsive forces lead to a non-smoothing process able to separate labels.

Submitted to 36th Conference on Neural Information Processing Systems (NeurIPS 2022). Do not distribute.

graph convolutional models always induce a low-frequency dominated dynamics *independent* of the sign and magnitude of the spectrum of the channel-mixing. We also discuss how simple choices make common architectures fit GRAFF and conduct thorough ablation studies to corroborate the theoretical analysis on the role of the spectrum of $\mathbf{W}$. (iii) We crystallize *an instance* of our framework into a linear, residual, convolutional model that achieves competitive performance on homophilic and heterophilic real world graphs whilst being faster than GCN.

**Related work.** Our analysis is related to studying GNNs as filters on the graph spectrum [15, 24, 2, 25] and over-smoothing [29, 30, 8, 50] and partly adopts techniques similar to [30]. The key difference is that we also consider the spectrum of the 'channel-mixing' matrix. The concept of gradient flows has been a standard tool in physics and geometry [16], from which they were adopted for image processing [26], and recently used in ML [35] for the analysis of Transformers [41] – see also [18] for discussion of loss landscapes. Our continuous-time evolution equations follows the spirit of Neural ODES [22, 12, 3] and the study of GNNs as continuous dynamical systems [44, 10, 17, 9].

**Outline.** In Section 2 we review the continuous and discrete Dirichlet energy and the associated gradient flow framework. We formalize the notion of over-smoothing and low(high)-frequency-dominated dynamics to investigate GNNs and study the dominant components in their evolution. We extend the graph Dirichlet energy to allow for a non-trivial norm for the feature edge-gradient. This leads to gradient flow equations that diffuse the features and over-smooth in the limit. Accordingly, in Section 3 we introduce a more general energy with a symmetric channel-mixing matrix $\mathbf{W}$ giving rise to attractive and repulsive pairwise terms via its positive and negative eigenvalues and show that the negative spectrum can induce high-frequency-dominant dynamics. In Section 4 we first compare with continuous GNN models and then discretize the equations and provide a 'recipe' for making standard GNN architectures fit a gradient flow framework. We adapt the spectral analysis to discrete-time showing that gradient flow convolutional models *can* generate a dynamics dominated by the high frequencies via the negative eigenvalues of $\mathbf{W}$ while this is impossible if we drop the residual connection. In Section 5 we corroborate our theoretical analysis on the role of the spectrum of $\mathbf{W}$ via ablation studies on graphs with varying homophily. Experiments on real world datasets show a competitive performance of our model despite its simplicity and reduced number of parameters.

## 2 Gradient-flow formalism

**Notations adopted throughout the paper.** Let $\mathsf{G} = (\mathsf{V}, \mathsf{E})$ be an *undirected* graph with $n$ nodes. We denote by $\mathbf{F} \in \mathbb{R}^{n \times d}$ the matrix of $d$-dimensional node features, by $\mathbf{f}_i \in \mathbb{R}^d$ its $i$-th row (transposed), by $\mathbf{f}^r \in \mathbb{R}^n$ its $r$-th column, and by $\mathrm{vec}(\mathbf{F}) \in \mathbb{R}^{nd}$ the vectorization of $\mathbf{F}$ obtained by stacking its columns. Given a symmetric matrix $\mathbf{B}$, we let $\lambda_+^{\mathbf{B}}, \lambda_-^{\mathbf{B}}$ denote its most positive and negative eigenvalues, respectively, and $\rho_{\mathbf{B}}$ be its *spectral radius*. If $\mathbf{B} \succeq 0$, then $\mathrm{gap}(\mathbf{B})$ denotes the *positive smallest eigenvalue* of $\mathbf{B}$. $\dot{f}(t)$ denotes the temporal derivative, $\otimes$ is the Kronecker product and 'a.e.' means *almost every* w.r.t. Lebesgue measure and usually refers to data in the complement of some lower dimensional subspace in $\mathbb{R}^{n \times d}$. Proofs and additional results appear in the Appendix.

**Starting point: a geometric parallelism.** To motivate a gradient-flow approach for GNNs, we start from the continuous case (see Appendix A.1 for details). Consider a smooth map $f : \mathbb{R}^n \to (\mathbb{R}^d, h)$ with $h$ a constant metric represented by $\mathbf{H} \succeq 0$. The *Dirichlet energy* of $f$ is defined by

$$\mathcal{E}(f, h) = \frac{1}{2} \int_{\mathbb{R}^n} \|\nabla f\|_h^2 \, dx = \frac{1}{2} \sum_{q,r=1}^d \sum_{j=1}^n \int_{\mathbb{R}^n} h_{qr} \partial_j f^q \partial_j f^r(x) dx \tag{1}$$

and measures the 'smoothness' of $f$. A natural approach to find minimizers of $\mathcal{E}$ - called *harmonic maps* - was introduced in [16] and consists in studying the **gradient flow** of $\mathcal{E}$, wherein a given map $f(0) = f_0$ is evolved according to $\dot{f}(t) = -\nabla_f \mathcal{E}(f(t))$. These type of evolution equations have historically been the core of *variational* and *PDE-based image processing*; in particular, gradient flows of the Dirichlet energy were shown [26] to recover the Perona-Malik nonlinear diffusion [32].

**Motivation: GNNs for node-classification.** We wish to extend the gradient flow formalism to node classification on graphs. Assume we have a graph $\mathsf{G}$, node-features $\mathbf{F}_0$ and labels $\{y_i\}$ on $\mathsf{V}_{\mathrm{train}} \subset \mathsf{V}$, and that we want to predict the labels on $\mathsf{V}_{\mathrm{test}} \subset \mathsf{V}$. A GNN typically evolves the features via some

87  parametric rule, $\text{GNN}_\theta(\mathsf{G}, \mathbf{F}_0)$, and uses a decoding map for the prediction $y = \psi_{\text{DE}}(\text{GNN}_\theta(\mathsf{G}, \mathbf{F}_0))$.
88  In graph convolutional models [15, 27], $\text{GNN}_\theta$ consists of two operations: applying a shared linear
89  transformation to the features (**'channel mixing'**) and propagating them along the edges of the graph
90  (**'diffusion'**). Our **goal** consists in studying when $\text{GNN}_\theta$ is the *gradient flow* of some parametric class
91  of energies $\mathcal{E}_\theta : \mathbb{R}^{n \times d} \to \mathbb{R}$, which generalize the Dirichlet energy. This means that the parameters
92  can be interpreted as 'finding the right notion of smoothness' for our task. We evolve the features by
93  $\dot{\mathbf{F}}(t) = -\nabla_{\mathbf{F}} \mathcal{E}_\theta(\mathbf{F}(t))$ with prediction $y = \psi_{\text{DE}}(\mathbf{F}(T))$ for some optimal time $T$.

94  **Why a gradient flow?** Since $\dot{\mathcal{E}}_\theta(\mathbf{F}(t)) = -||\nabla_{\mathbf{F}} \mathcal{E}_\theta(\mathbf{F}(t))||^2$, the energy dissipates along the gradient
95  flow. Accordingly, this framework allows to *explain the GNN dynamics* as flowing the node features
96  in the direction of steepest descent of $\mathcal{E}_\theta$. Indeed, we find that parametrizing an energy leads to
97  equations governed by attractive and repulsive forces that can be controlled via the spectrum of
98  symmetric 'channel-mixing' matrices. This shows that by learning to distribute more mass over the
99  negative (positive) eigenvalues of the channel-mixing, gradient flow models can generate dynamics
100  dominated by the higher (respectively, lower) graph frequencies and hence tackle different homophily
101  scenarios. The gradient flow framework also leads to sharing of the weights across layers (since we
102  parametrize the *energy* rather than the *evolution equations*, as usually done in GNNs), allowing us to
103  reduce the number of parameters without compromising performance (see Table 1).

104  **Analysis on graphs: preliminaries.**   Given a *connected* graph $\mathsf{G}$ with self-loops, its adjacency
105  matrix $\mathbf{A}$ is defined as $a_{ij} = 1$ if $(i, j) \in \mathsf{E}$ and zero otherwise. We let $\mathbf{D} = \text{diag}(d_i)$ be the degree
106  matrix and write $\bar{\mathbf{A}} := \mathbf{D}^{-1/2} \mathbf{A} \mathbf{D}^{-1/2}$. Let $\mathbf{F} \in \mathbb{R}^{n \times d}$ be the matrix representation of a signal. Its
107  *graph gradient* is $(\nabla \mathbf{F})_{ij} := \mathbf{f}_j/\sqrt{d_j} - \mathbf{f}_i/\sqrt{d_i}$. We define the *Laplacian* as $\mathbf{\Delta} := -\frac{1}{2}\text{div}\,\nabla$ (the
108  *divergence* div is the adjoint of $\nabla$), represented by $\mathbf{\Delta} = \mathbf{I} - \bar{\mathbf{A}} \succeq 0$. We refer to the eigenvalues of
109  $\mathbf{\Delta}$ as *frequencies*: the lowest frequency is always 0 while the highest frequency is $\rho_{\mathbf{\Delta}} \leq 2$ [14]. As
110  for the continuum case, the gradient allows to define a *(graph) Dirichlet energy* as [49]

$$\mathcal{E}^{\text{Dir}}(\mathbf{F}) := \frac{1}{4} \sum_i \sum_{j:(i,j)\in\mathsf{E}} ||(\nabla\mathbf{F})_{ij}||^2 \equiv \frac{1}{4} \sum_{(i,j)\in\mathsf{E}} ||\frac{\mathbf{f}_i}{\sqrt{d_i}} - \frac{\mathbf{f}_j}{\sqrt{d_j}}||^2 = \frac{1}{2}\text{trace}(\mathbf{F}^\top \mathbf{\Delta} \mathbf{F}), \quad (2)$$

111  where the extra $\frac{1}{2}$ is for convenience. As for manifolds, $\mathcal{E}^{\text{Dir}}$ measures smoothness. If we stack the
112  columns of $\mathbf{F}$ into $\text{vec}(\mathbf{F}) \in \mathbb{R}^{nd}$, the gradient flow of $\mathcal{E}^{\text{Dir}}$ yields the *heat equation* on each channel:

$$\text{vec}(\dot{\mathbf{F}}(t)) = -\nabla_{\text{vec}(\mathbf{F})} \mathcal{E}^{\text{Dir}}(\text{vec}(\mathbf{F}(t))) = -(\mathbf{I}_d \otimes \mathbf{\Delta})\text{vec}(\mathbf{F}(t)) \iff \dot{\mathbf{f}}^r(t) = -\mathbf{\Delta}\mathbf{f}^r(t), \quad (3)$$

113  for $1 \leq r \leq d$. Similarly to [8], we rely on $\mathcal{E}^{\text{Dir}}$ to assess whether a given dynamics $t \mapsto \mathbf{F}(t)$ is a
114  smoothing process. A different choice of Laplacian $\mathbf{L} = \mathbf{D} - \mathbf{A}$ with non-normalized adjacency
115  induces the analogous Dirichlet energy $\mathcal{E}^{\text{Dir}}_{\mathbf{L}}(\mathbf{F}) = \frac{1}{2}\text{trace}(\mathbf{F}^\top \mathbf{L} \mathbf{F})$. Throughout this paper, we rely
116  on the following definitions (see Appendix A.3 for further equivalent formulations and justifications):
117  **Definition 2.1.** $\dot{\mathbf{F}}(t) = \text{GNN}_\theta(\mathbf{F}(t), t)$ initialized at $\mathbf{F}(0)$ is *smoothing* if $\mathcal{E}^{\text{Dir}}(\mathbf{F}(t)) \leq C + \varphi(t)$,
118  with $C$ a constant only depending on $\mathcal{E}^{\text{Dir}}(\mathbf{F}(0))$ and $\dot{\varphi}(t) \leq 0$. *Over-smoothing* occurs if either
119  $\mathcal{E}^{\text{Dir}}(\mathbf{F}(t)) \to 0$ or $\mathcal{E}^{\text{Dir}}_{\mathbf{L}}(\mathbf{F}(t)) \to 0$ for $t \to \infty$.

120  Our notion of 'over-smoothing' is a relaxed version of the definition in [34] – although in the linear
121  case one always finds an *exponential decay* of $\mathcal{E}^{\text{Dir}}$. We note that $\mathcal{E}^{\text{Dir}}(\mathbf{F}(t)) \to 0$ iff $\mathbf{\Delta}\mathbf{f}^r(t) \to \mathbf{0}$ for
122  each column $\mathbf{f}^r$. As in [30], this corresponds to a loss of separation power along the solution where
123  nodes with *equal degree* become indistinguishable since we converge to $\ker(\mathbf{\Delta})$ (if we replaced $\mathbf{\Delta}$
124  with $\mathbf{L}$ then we would not even be able to separate nodes with different degrees in the limit).

125  To motivate the next definition, consider $\dot{\mathbf{F}}(t) = \bar{\mathbf{A}}\mathbf{F}(t)$. Despite $||\mathbf{F}(t)||$ being unbounded for a.e.
126  $\mathbf{F}(0)$, the low-frequency components are growing the fastest and indeed $\mathbf{F}(t)/||\mathbf{F}(t)|| \to \mathbf{F}_\infty$ s.t.
127  $\mathbf{\Delta}\mathbf{f}^r_\infty = \mathbf{0}$ for $1 \leq r \leq d$. We formalize this scenario – including the opposite case of high-frequency
128  components being dominant – by studying $\mathcal{E}^{\text{Dir}}(\mathbf{F}(t)/||\mathbf{F}(t)||)$, i.e. the Rayleigh quotient of $\mathbf{I}_d \otimes \mathbf{\Delta}$.
129  **Definition 2.2.** $\dot{\mathbf{F}}(t) = \text{GNN}_\theta(\mathbf{F}(t), t)$ initialized at $\mathbf{F}(0)$ is *Low/High-Frequency-Dominant*
130  (L/HFD) if $\mathcal{E}^{\text{Dir}}(\mathbf{F}(t)/||\mathbf{F}(t)||) \to 0$ (respectively, $\mathcal{E}^{\text{Dir}}(\mathbf{F}(t)/||\mathbf{F}(t)||) \to \rho_{\mathbf{\Delta}}/2$) for $t \to \infty$.

131  We report a consequence of Definition 2.2 and refer to Appendix A.3 for additional details and
132  motivations for the characterizations of LFD and HFD.
133  **Lemma 2.3.** $\text{GNN}_\theta$ *is LFD (HFD) iff for each* $t_j \to \infty$ *there exist* $t_{j_k} \to \infty$ *and* $\mathbf{F}_\infty$ *s.t.*
134  $\mathbf{F}(t_{j_k})/||\mathbf{F}(t_{j_k})|| \to \mathbf{F}_\infty$ *and* $\mathbf{\Delta}\mathbf{f}^r_\infty = \mathbf{0}$ *(* $\mathbf{\Delta}\mathbf{f}^r_\infty = \rho_{\mathbf{\Delta}}\mathbf{f}^r_\infty$*, respectively).*

If a graph is *homophilic*, adjacent nodes are likely to share the same label and we expect a smoothing or LFD dynamics enhancing the low-frequency components to be successful at node classification tasks [43, 28]. In the opposite case of *heterophily*, the high-frequency components might contain more relevant information for separating classes [4, 5] – the prototypical example being the eigenvector of $\mathbf{\Delta}$ associated with largest frequency $\rho_{\mathbf{\Delta}}$ separating a regular bipartite graph. In other words, the class of heterophilic graphs contain instances where signals should be *sharpened* by increasing $\mathcal{E}^{\mathrm{Dir}}$ rather than smoothed out. Accordingly, an ideal framework for learning on graphs must accommodate both of these opposite scenarios by being able to induce either an LFD or a HFD dynamics.

**Parametric Dirichlet energy: channel-mixing as metric in feature space.** In eq. (1) a constant nontrivial metric $h$ in $\mathbb{R}^d$ leads to the mixing of the feature channels. We adapt this idea by considering a symmetric positive semi-definite $\mathbf{H} = \mathbf{W}^{\top}\mathbf{W}$ with $\mathbf{W} \in \mathbb{R}^{d \times d}$ and using it to generalize $\mathcal{E}^{\mathrm{Dir}}$ as

$$\mathcal{E}^{\mathrm{Dir}}_{\mathbf{W}}(\mathbf{F}) := \frac{1}{4}\sum_{q,r=1}^{d}\sum_{i}\sum_{j:(i,j)\in \mathsf{E}} h_{qr}(\nabla \mathbf{f}^q)_{ij}(\nabla \mathbf{f}^r)_{ij} = \frac{1}{4}\sum_{(i,j)\in \mathsf{E}}||\mathbf{W}(\nabla \mathbf{F})_{ij}||^2. \qquad (4)$$

We note the analogy with eq. (1), where the sum over the nodes replaces the integration over the domain and the $j$-th derivative at some point $i$ is replaced by the gradient along the edge $(i,j) \in \mathsf{E}$. We generally treat $\mathbf{W}$ as *learnable weights* and study the gradient flow of $\mathcal{E}^{\mathrm{Dir}}_{\mathbf{W}}$:

$$\dot{\mathbf{F}}(t) = -\nabla_{\mathbf{F}}\mathcal{E}^{\mathrm{Dir}}_{\mathbf{W}}(\mathbf{F}(t)) = -\mathbf{\Delta}\mathbf{F}(t)\mathbf{W}^{\top}\mathbf{W}. \qquad (5)$$

We see that eq. (5) generalizes eq. (3). Below 'smoothing' is intended as in Definition 2.1.

**Proposition 2.4.** *Let $P^{\mathrm{ker}}_{\mathbf{W}}$ be the projection onto $\ker(\mathbf{W}^{\top}\mathbf{W})$. Equation (5) is smoothing since*

$$\mathcal{E}^{\mathrm{Dir}}(\mathbf{F}(t)) \leq e^{-2t\mathrm{gap}(\mathbf{W}^{\top}\mathbf{W})\mathrm{gap}(\mathbf{\Delta})}||\mathbf{F}(0)||^2 + \mathcal{E}^{\mathrm{Dir}}((P^{\mathrm{ker}}_{\mathbf{W}}\otimes \mathbf{I}_n)\mathrm{vec}(\mathbf{F}(0))), \quad t \geq 0.$$

*In fact $\mathbf{F}(t) \to \mathbf{F}_{\infty}$ s.t. $\exists\, \boldsymbol{\phi}_{\infty} \in \mathbb{R}^d$: for each $i \in \mathsf{V}$ we have $(\mathbf{f}_{\infty})_i = \sqrt{d_i}\boldsymbol{\phi}_{\infty} + P^{\mathrm{ker}}_{\mathbf{W}}\mathbf{f}_i(0)$.*

Proposition 2.4 implies that *no weight matrix $\mathbf{W}$ in eq. (5) can separate the limit embeddings $\mathbf{F}(\infty)$ of nodes with same degree and input features*. If $\mathbf{W}$ has a trivial kernel, then nodes with same degrees converge to the same representation and *over-smoothing* occurs as per Definition 2.1. Differently from [29, 30, 8], over-smoothing occurs independently of the spectral radius of the 'channel-mixing' if its eigenvalues are *positive* – even for equations which lead to residual GNNs when discretized [12]. According to Proposition 2.4, we do not expect eq. (5) to succeed on heterophilic graphs where *smoothing* processes are generally harmful – this is confirmed in Figure 2 (see *prod*-curve). To remedy this problem, we generalize eq. (5) to a gradient flow that can be HFD as per Definition 2.2.

## 3 A general parametric energy for pairwise interactions

We first rewrite the energy $\mathcal{E}^{\mathrm{Dir}}_{\mathbf{W}}$ in eq. (4) as

$$\mathcal{E}^{\mathrm{Dir}}_{\mathbf{W}}(\mathbf{F}) = \frac{1}{2}\sum_{i}\langle \mathbf{f}_i, \mathbf{W}^{\top}\mathbf{W}\mathbf{f}_i\rangle - \frac{1}{2}\sum_{i,j}\bar{a}_{ij}\langle \mathbf{f}_i, \mathbf{W}^{\top}\mathbf{W}\mathbf{f}_j\rangle. \qquad (6)$$

We then define a *new, more general* energy by replacing the occurrences of $\mathbf{W}^{\top}\mathbf{W}$ with new symmetric matrices $\mathbf{\Omega}, \mathbf{W} \in \mathbb{R}^{d \times d}$ since we also want to generate repulsive forces:

$$\mathcal{E}^{\mathrm{tot}}(\mathbf{F}) := \frac{1}{2}\sum_{i}\langle \mathbf{f}_i, \mathbf{\Omega}\mathbf{f}_i\rangle - \frac{1}{2}\sum_{i,j}\bar{a}_{ij}\langle \mathbf{f}_i, \mathbf{W}\mathbf{f}_j\rangle \equiv \mathcal{E}^{\mathrm{ext}}_{\mathbf{\Omega}}(\mathbf{F}) + \mathcal{E}^{\mathrm{pair}}_{\mathbf{W}}(\mathbf{F}), \qquad (7)$$

with associated gradient flow of the form (see Appendix B)

$$\dot{\mathbf{F}}(t) = -\nabla_{\mathbf{F}}\mathcal{E}^{\mathrm{tot}}(\mathbf{F}(t)) = -\mathbf{F}(t)\mathbf{\Omega} + \bar{\mathbf{A}}\mathbf{F}(t)\mathbf{W}. \qquad (8)$$

Note that eq. (8) *is gradient flow of some energy $\mathbf{F} \mapsto \mathcal{E}^{\mathrm{tot}}(\mathbf{F})$ iff both $\mathbf{\Omega}$ and $\mathbf{W}$ are symmetric.*

**A multi-particle system point of view: attraction vs repulsion.** Consider the $d$-dimensional node-features as particles in $\mathbb{R}^d$ with energy $\mathcal{E}^{\mathrm{tot}}$. While the term $\mathcal{E}^{\mathrm{ext}}_{\mathbf{\Omega}}$ is *independent of the graph topology* and represents an **external** field in the feature space, the second term $\mathcal{E}^{\mathrm{pair}}_{\mathbf{W}}$ constitutes a potential energy, with $\mathbf{W}$ a *bilinear form* determining the **pairwise interactions** of adjacent node

representations. Given a symmetric $\mathbf{W}$, we write $\mathbf{W} = \mathbf{\Theta}_+^\top \mathbf{\Theta}_+ - \mathbf{\Theta}_-^\top \mathbf{\Theta}_-$, by decomposing the spectrum of $\mathbf{W}$ in positive and negative values. We can rewrite $\mathcal{E}^{\text{tot}} = \mathcal{E}^{\text{ext}}_{\mathbf{\Omega}-\mathbf{W}} + \mathcal{E}^{\text{Dir}}_{\mathbf{\Theta}_+} - \mathcal{E}^{\text{Dir}}_{\mathbf{\Theta}_-}$, i.e.

$$\mathcal{E}^{\text{tot}}(\mathbf{F}) = \frac{1}{2}\sum_i \langle \mathbf{f}_i, (\mathbf{\Omega} - \mathbf{W})\mathbf{f}_i \rangle + \frac{1}{4}\sum_{i,j}||\mathbf{\Theta}_+(\nabla\mathbf{F})_{ij}||^2 - \frac{1}{4}\sum_{i,j}||\mathbf{\Theta}_-(\nabla\mathbf{F})_{ij}||^2. \quad (9)$$

The gradient flow of $\mathcal{E}^{\text{tot}}$ *minimizes* $\mathcal{E}^{\text{Dir}}_{\mathbf{\Theta}_+}$ and *maximizes* $\mathcal{E}^{\text{Dir}}_{\mathbf{\Theta}_-}$. The matrix $\mathbf{W}$ encodes *repulsive pairwise interactions* via its negative-definite component $\mathbf{\Theta}_-$ which lead to terms $||\mathbf{\Theta}_-(\nabla\mathbf{F})_{ij}||$ increasing along the solution. The latter affords a 'sharpening' effect desirable on heterophilic graphs where we need to disentangle adjacent node representations and hence 'magnify' the edge-gradient.

**Spectral analysis of the channel-mixing.** We will now show that eq. (8) can lead to a HFD dynamics. To this end, we assume that $\mathbf{\Omega} = \mathbf{0}$ so that eq. (8) becomes $\dot{\mathbf{F}}(t) = \bar{\mathbf{A}}\mathbf{F}(t)\mathbf{W}$. According to eq. (9) the negative eigenvalues of $\mathbf{W}$ lead to repulsion. We show that the latter can induce HFD dynamics as per Definition 2.2. We let $P^{\rho_-}_{\mathbf{W}}$ be the orthogonal projection into the eigenspace of $\mathbf{W} \otimes \bar{\mathbf{A}}$ associated with the eigenvalue $\rho_- := |\lambda^{\mathbf{W}}_-|(\rho_{\mathbf{\Delta}} - 1)$. We define $\epsilon_{\text{HFD}}$ explicitly in eq. (24).

**Proposition 3.1.** *If $\rho_- > \lambda^{\mathbf{W}}_+$, then $\dot{\mathbf{F}}(t) = \bar{\mathbf{A}}\mathbf{F}(t)\mathbf{W}$ is HFD for a.e. $\mathbf{F}(0)$: there exists $\epsilon_{\text{HFD}}$ s.t.*

$$\mathcal{E}^{\text{Dir}}(\mathbf{F}(t)) = e^{2t\rho_-}\left(\frac{\rho_{\mathbf{\Delta}}}{2}||P^{\rho_-}_{\mathbf{W}}\mathbf{F}(0)||^2 + \mathcal{O}(e^{-2t\epsilon_{\text{HFD}}})\right), \quad t \geq 0,$$

*and $\mathbf{F}(t)/||\mathbf{F}(t)||$ converges to $\mathbf{F}_\infty \in \mathbb{R}^{n \times d}$ such that $\mathbf{\Delta}\mathbf{f}^r_\infty = \rho_{\mathbf{\Delta}}\mathbf{f}^r_\infty$, for $1 \leq r \leq d$.*

Proposition 3.1 shows that *if enough mass of the spectrum of the 'channel-mixing' is distributed over the negative eigenvalues, then the evolution is dominated by the graph high frequencies.* This analysis is made possible in our gradient flow framework where $\mathbf{W}$ must be *symmetric*. The HFD dynamics induced by negative eigenvalues of $\mathbf{W}$ is confirmed in Figure 2 (*neg-prod*-curve in the bottom chart).

**A more general energy.** Equations with a source term may have better expressive power [44, 11, 39]. In our framework this means adding an extra energy term of the form $\mathcal{E}^{\text{source}}_{\tilde{\mathbf{W}}}(\mathbf{F}) := \beta\langle\mathbf{F}, \mathbf{F}(0)\tilde{\mathbf{W}}\rangle$ to eq. (7) with some learnable $\beta$ and $\tilde{\mathbf{W}}$. This leads to the following gradient flow:

$$\dot{\mathbf{F}}(t) = -\mathbf{F}(t)\mathbf{\Omega} + \bar{\mathbf{A}}\mathbf{F}(t)\mathbf{W} - \beta\mathbf{F}(0)\tilde{\mathbf{W}}. \quad (10)$$

We also observe that one could replace the fixed matrix $\bar{\mathbf{A}}$ with a more general *symmetric graph vector field* $\mathcal{A}$ satisfying $\mathcal{A}_{ij} = 0$ if $(i, j) \notin \mathsf{E}$, although in this work we focus on the case $\mathcal{A} = \bar{\mathbf{A}}$. We also note that when $\mathbf{\Omega} = \mathbf{W}$, then eq. (8) becomes $\dot{\mathbf{F}}(t) = -\mathbf{\Delta}\mathbf{F}(t)\mathbf{W}$. We perform a spectral analysis of this case in Appendix B.2.

**Non-linear activations.** In Appendix B.3 we discuss non-linear gradient flow equations. Here we study what happens if the gradient flow in eq. (10) is activated *pointwise* by $\sigma : \mathbb{R} \to \mathbb{R}$. We show that although we are no longer a gradient flow, the learnable multi-particle energy $\mathcal{E}^{\text{tot}}$ is still decreasing along the solution, meaning that the interpretation of the channel-mixing $\mathbf{W}$ inducing attraction and repulsion via its positive and negative eigenvalues respectively **is preserved**.

**Proposition 3.2.** *Consider a non-linear map $\sigma : \mathbb{R} \to \mathbb{R}$ such that the function $x \mapsto x\sigma(x) \geq 0$. If $t \mapsto \mathbf{F}(t)$ solves the equation*

$$\dot{\mathbf{F}}(t) = \sigma\left(-\mathbf{F}(t)\mathbf{\Omega} + \bar{\mathbf{A}}\mathbf{F}(t)\mathbf{W} - \beta\mathbf{F}(0)\tilde{\mathbf{W}}\right),$$

*where $\sigma$ acts elementwise, then*

$$\frac{d\mathcal{E}^{\text{tot}}(\mathbf{F}(t))}{dt} \leq 0.$$

A proof of this result and more details and discussion are reported in Appendix E. We emphasize here that differently from previous results about behaviour of ReLU wrt $\mathcal{E}^{\text{Dir}}$ [30, 8], we deal with a much more general energy that can also induce repulsion and a more general family of activation functions (that include ReLU, tanh, arctan and many others).

## 4 Comparison with GNNs

In this Section, we study standard GNN models from the perspective of our gradient flow framework.

### 4.1 Continuous case

Continuous GNN models replace layers with continuous time. In contrast with Proposition 3.1 we show that three main *linearized* continuous GNN models are either *smoothing* or LFD as per Definition 2.2. The linearized PDE-GCN$_D$ model [17] corresponds to choosing $\beta = 0$ and $\mathbf{\Omega} = \mathbf{W} = \mathbf{K}(t)^\top \mathbf{K}(t)$ in eq. (10), for some time-dependent family $t \mapsto \mathbf{K}(t) \in \mathbb{R}^{d \times d}$:

$$\dot{\mathbf{F}}_{\mathrm{PDE-GCN_D}}(t) = -\mathbf{\Delta}\mathbf{F}(t)\mathbf{K}(t)^\top \mathbf{K}(t).$$

The CGNN model [44] can be derived from eq. (10) by setting $\mathbf{\Omega} = \mathbf{I} - \tilde{\mathbf{\Omega}}, \mathbf{W} = \tilde{\mathbf{W}} = \mathbf{I}, \beta = 1$:

$$\dot{\mathbf{F}}_{\mathrm{CGNN}}(t) = -\mathbf{\Delta}\mathbf{F}(t) + \mathbf{F}(t)\tilde{\mathbf{\Omega}} + \mathbf{F}(0).$$

Finally, in linearized GRAND [10] a row-stochastic matrix $\mathcal{A}(\mathbf{F}(0))$ is *learned* from the encoding via an attention mechanism and we have

$$\dot{\mathbf{F}}_{\mathrm{GRAND}}(t) = -\mathbf{\Delta}_{\mathrm{RW}}\mathbf{F}(t) = -(\mathbf{I} - \mathcal{A}(\mathbf{F}(0)))\mathbf{F}(t).$$

We note that if $\mathcal{A}$ is not symmetric, then GRAND is *not* a gradient flow.

**Proposition 4.1.** $\mathrm{PDE} - \mathrm{GCN}_D$, CGNN *and* GRAND *satisfy the following:*

*(i)* $\mathrm{PDE} - \mathrm{GCN}_D$ *is a smoothing model:* $\dot{\mathcal{E}}^{\mathrm{Dir}}(\mathbf{F}_{\mathrm{PDE-GCN_D}}(t)) \leq 0$.

*(ii) For a.e.* $\mathbf{F}(0)$ *it holds:* CGNN *is never* HFD *and if we remove the source term, then* $\mathcal{E}^{\mathrm{Dir}}(\mathbf{F}_{\mathrm{CGNN}}(t)/||\mathbf{F}_{\mathrm{CGNN}}(t)||) \leq e^{-\mathrm{gap}(\mathbf{\Delta})t}$.

*(iii) If* G *is connected,* $\mathbf{F}_{\mathrm{GRAND}}(t) \to \boldsymbol{\mu}$ *as* $t \to \infty$, *with* $\boldsymbol{\mu}^r = \mathrm{mean}(\mathbf{f}^r(0))$, $1 \leq r \leq d$.

By (ii) the source-free CGNN-evolution is LFD *independent of* $\tilde{\mathbf{\Omega}}$. Moreover, by (iii), over-smoothing occurs for GRAND as per Definition 2.1. On the other hand, Proposition 3.1 shows that the negative eigenvalues of $\mathbf{W}$ can make the source-free gradient flow in eq. (8) HFD. Experiments in Section 5 confirm that the gradient flow model outperforms CGNN and GRAND on heterophilic graphs.

### 4.2 Discrete case

We now describe a discrete version of our gradient flow model and compare it to 'discrete' GNNs where discrete time steps correspond to different layers. In the spirit of [12], we use explicit Euler scheme with step size $\tau \leq 1$ to solve eq. (10) and set $\tilde{\mathbf{W}} = \mathbf{I}$. In the gradient flow framework we *parametrize the energy* rather than the actual equations, which leads to *symmetric* channel-mixing matrices $\mathbf{\Omega}, \mathbf{W} \in \mathbb{R}^{d \times d}$ that are *shared across the layers*. Since the matrices are square, an *encoding* block $\psi_{\mathrm{EN}} : \mathbb{R}^{n \times p} \to \mathbb{R}^{n \times d}$ is used to process input features $\mathbf{F}_0 \in \mathbb{R}^{n \times p}$ and generally reduce the hidden dimension from $p$ to $d$. Moreover, the iterations inherently lead to a residual architecture because of the explicit Euler discretization:

$$\mathbf{F}(t + \tau) = \mathbf{F}(t) + \tau\left(-\mathbf{F}(t)\mathbf{\Omega} + \bar{\mathbf{A}}\mathbf{F}(t)\mathbf{W} + \beta\mathbf{F}(0)\right), \quad \mathbf{F}(0) = \psi_{\mathrm{EN}}(\mathbf{F}_0), \qquad (11)$$

with prediction $y = \psi_{\mathrm{DE}}(\mathbf{F}(T))$ produced by a *decoder* $\psi_{\mathrm{DE}} : \mathbb{R}^{n \times d} \to \mathbb{R}^{n \times k}$, where $k$ is the number of label classes and $T$ *integration time* of the form $T = m\tau$, so that $m \in \mathbb{N}$ represents the number of *layers*. Although eq. (11) is linear, we can include non-linear activations in $\psi_{\mathrm{EN}}, \psi_{\mathrm{DE}}$ making the entire model generally non-linear. We emphasize two important points:

- Since the framework is residual, even if the message-passing is linear, this is *not equivalent* to collapsing the dynamics into a single layer with diffusion matrix $\bar{\mathbf{A}}^m$, with $m$ the number of layers, see eq. (27) in the appendix where we derive the expansion of the solution.

- We could also activate the equations pointwise and maintain the physics interpretation thanks to Proposition 3.2 to gain greater expressive power. In the following though, we mainly stick to the linear discrete gradient flow unless otherwise stated.

**Are discrete GNNs gradient flows?** Given a (learned) symmetric graph vector field $\mathcal{A} \in \mathbb{R}^{n \times n}$ satisfying $\mathcal{A}_{ij} = 0$ if $(i, j) \notin \mathsf{E}$, consider a family of linear GNNs with shared weights of the form

$$\mathbf{F}(t + 1) = \mathbf{F}(t)\mathbf{\Omega} + \mathcal{A}\mathbf{F}(t)\mathbf{W} + \beta\mathbf{F}(0)\tilde{\mathbf{W}}, \quad 0 \leq t \leq T. \qquad (12)$$

Symmetry is the key requirement to interpret GNNs in eq. (12) in a gradient flow framework.

**Lemma 4.2.** *Equation* (12) *is the unit step size discrete gradient flow of $\mathcal{E}_{\mathbf{I}-\Omega}^{\mathrm{ext}} + \mathcal{E}_{\boldsymbol{\mathcal{A}},\mathbf{W}}^{\mathrm{pair}} - \mathcal{E}_{\tilde{\mathbf{W}}}^{\mathrm{source}}$, with $\mathcal{E}_{\boldsymbol{\mathcal{A}},\mathbf{W}}^{\mathrm{pair}}$ defined by replacing $\bar{\mathbf{A}}$ with $\boldsymbol{\mathcal{A}}$ in eq.* (7), *iff $\Omega$ and $\mathbf{W}$ are symmetric.*

Lemma 4.2 provides a recipe for making standard architectures into a gradient flow, with *symmetry* being the key requirement. When eq. (12) is a gradient flow, the underlying GNN dynamics is equivalent to minimizing a multi-particle energy by learning attractive and repulsive directions in feature space as discussed in Section 3. In Appendix C.2, we show how Lemma 4.2 covers linear versions of GCN [27, 43], GAT [42], GraphSAGE [23] and GCNII [11] to name a few.

**Over-smoothing analysis in discrete setting.** By Proposition 3.1 we know that the continuous version of eq. (11) can be HFD thanks to the negative eigenvalues of $\mathbf{W}$. The next result represents a discrete counterpart of Proposition 3.1 and shows that *residual, symmetrized graph convolutional models can be* HFD. Below $P_{\mathbf{W}}^{\rho_-}$ is the projection into the eigenspace associated with the eigenvalue $\rho_- := |\lambda_-^{\mathbf{W}}|(\rho_{\boldsymbol{\Delta}} - 1)$ and we report the explicit value of $\delta_{\mathrm{HFD}}$ in eq. (28) in Appendix C.3. We let:

$$\lambda_+^{\mathbf{W}}(\rho_{\boldsymbol{\Delta}} - 1))^{-1} < |\lambda_-^{\mathbf{W}}| < 2(\tau(2 - \rho_{\boldsymbol{\Delta}}))^{-1}. \tag{13}$$

**Theorem 4.3.** *Given $\mathbf{F}(t + \tau) = \mathbf{F}(t) + \tau\bar{\mathbf{A}}\mathbf{F}(t)\mathbf{W}$, with $\mathbf{W}$ symmetric, if eq.* (13) *holds then*

$$\mathcal{E}^{\mathrm{Dir}}(\mathbf{F}(m\tau)) = (1 + \tau\rho_-)^{2m}\left(\frac{\rho_{\boldsymbol{\Delta}}}{2}||P_{\mathbf{W}}^{\rho_-}\mathbf{F}(0)||^2 + \mathcal{O}\left(\left(\frac{1 + \tau\delta_{\mathrm{HFD}}}{1 + \tau\rho_-}\right)^{2m}\right)\right), \quad \delta_{\mathrm{HFD}} < \rho_-,$$

*hence the dynamics is* HFD *for a.e. $\mathbf{F}(0)$ and in fact $\mathbf{F}(m\tau)/||\mathbf{F}(m\tau)|| \to \mathbf{F}_\infty$ s.t. $\boldsymbol{\Delta}\mathbf{f}_\infty^r = \rho_{\boldsymbol{\Delta}}\mathbf{f}_\infty^r$. Conversely, if G is not bipartite, then for a.e. $\mathbf{F}(0)$ the system $\mathbf{F}(t + \tau) = \tau\bar{\mathbf{A}}\mathbf{F}(t)\mathbf{W}$, with $\mathbf{W}$ symmetric, is* LFD *independent of the spectrum of $\mathbf{W}$.*

Theorem 4.3 shows that linear discrete gradient flows can be HFD due to the negative eigenvalues of $\mathbf{W}$. This differs from statements that standard GCNs act as low-pass filters and thus over-smooth in the limit. Indeed, in these cases the spectrum of $\mathbf{W}$ is generally ignored [43, 11] or required to be sufficiently small in terms of singular value decomposition [29, 30, 8] *when no residual connection is present*. On the other hand, Theorem 4.3 emphasizes that the spectrum of $\mathbf{W}$ plays a key role to enhance the high frequencies when enough mass is distributed over the negative eigenvalues provided that a residual connection exists – this is confirmed by the *neg-prod*-curve in Figure 2.

**The residual connection from a spectral perspective.** Given a sufficiently small step-size so that the right hand side of inequality 13 is satisfied, $\mathbf{F}(t + \tau) = \mathbf{F}(t) + \tau\bar{\mathbf{A}}\mathbf{F}(t)\mathbf{W}$ is HFD for a.e. $\mathbf{F}(0)$ if $|\lambda_-^{\mathbf{W}}|(\rho_{\boldsymbol{\Delta}} - 1) > \lambda_+^{\mathbf{W}}$, i.e. 'there is more mass' in the negative spectrum of $\mathbf{W}$ than in the positive one. This means that differently from [29, 30, 8], there is no requirement on the minimal magnitude of the spectral radius of $\mathbf{W}$ coming from the graph topology as long as $\lambda_+^{\mathbf{W}}$ is small enough. Conversely, without a residual term, the dynamics is LFD for a.e. $\mathbf{F}(0)$ *independently* of the sign and magnitude of the eigenvalues of $\mathbf{W}$. This is also confirmed by the GCN-curve in Figure 2.

**Over-smoothing vs** LFD**.** We highlight how in general a linear GCN equation as $\mathbf{F}(t + \tau) = \tau\bar{\mathbf{A}}\mathbf{F}(t)\mathbf{W}$ may avoid over-smoothing in the sense of Definition 2.1 meaning that $\mathcal{E}^{\mathrm{Dir}}(\mathbf{F}(t)) \to \infty$ as soon as there exist $\lambda_i^{\boldsymbol{\Delta}} \in (0, 1)$ and the spectral radius of $\mathbf{W}$ is large enough. However, this will not lead to over-separation since the dominating term is the lowest frequency one: in other words, once we re-set the scale right as per the normalization in Theorem 4.3, we encounter loss of separability even with large (and possibly negative) spectrum of $\mathbf{W}$.

## 5 Experiments

In this section we evaluate the gradient flow framework (GRAFF). We corroborate the spectral analysis using synthetic data with controllable homophily. We confirm that having negative (positive) eigenvalues of the channel-mixing $\mathbf{W}$ are essential in heterophilic (homophilic) scenarios where the gradient flow should align with HFD (LFD) respectively. We show that the gradient flow in eq. (11) – a linear, residual, symmetric graph convolutional model – achieves competitive performance on heterophilic datasets.

**Methodology.** We crystallize GRAFF in the model presented in eq. (11) with $\psi_{\text{EN}}, \psi_{\text{DE}}$ implemented as single linear layers or MLPs, and we set $\mathbf{\Omega}$ to be diagonal. For the real-world experiments we consider *diagonally-dominant* (DD), *diagonal* (D) and *time-dependent* choices for the structure of $\mathbf{W}$ that offer explicit control over its spectrum. In the (DD)-case, we consider a $\mathbf{W}^0 \in \mathbb{R}^{d \times d}$ *symmetric* with zero diagonal and $\mathbf{w} \in \mathbb{R}^d$ defined by $\mathbf{w}_\alpha = q_\alpha \sum_\beta |\mathbf{W}^0_{\alpha\beta}| + r_\alpha$, and set $\mathbf{W} = \text{diag}(\mathbf{w}) + \mathbf{W}^0$. Due to the Gershgorin Theorem the eigenvalues of $\mathbf{W}$ belong to $[\mathbf{w}_\alpha - \sum_\beta |\mathbf{W}^0_{\alpha\beta}|, \mathbf{w}_\alpha + \sum_\beta |\mathbf{W}^0_{\alpha\beta}|]$, so the model 'can' easily re-distribute mass in the spectrum of $\mathbf{W}$ via $q_\alpha, r_\alpha$. This generalizes the decomposition of $\mathbf{W}$ in [11] providing a justification in terms of its spectrum and turns out to be more efficient w.r.t. the hidden dimension $d$ as shown in Figure 4 in the Appendix. For (D) we take $\mathbf{W}$ to be diagonal, with entries sampled $\mathcal{U}[-1, 1]$ and fixed – i.e., **we do not train** over $\mathbf{W}$ – and only learn $\psi_{\text{EN}}, \psi_{\text{DE}}$. We also include a *time-dependent* model where $\mathbf{W}_t$ varies across layers. To investigate the role of the spectrum of $\mathbf{W}$ on synthetic graphs, we construct three additional variants: $\mathbf{W} = \mathbf{W}' + \mathbf{W}'^\top$, $\mathbf{W} = \pm \mathbf{W}'^\top \mathbf{W}'$ named *sum*, *prod* and *neg-prod* respectively where *prod* (*neg-prod*) variants have only non-negative (non-positive) eigenvalues.

**Complexity and number of parameters.** If we treat the number of layers as a constant, the discrete gradient flow scales as $\mathcal{O}(|\mathsf{V}|pd + |\mathsf{E}|d^2)$, where $p$ and $d$ are input feature and hidden dimension respectively, with $p \geq d$ usually. Note that GCN has complexity $\mathcal{O}(|\mathsf{E}|pd)$ and in fact *our model is faster than GCN* as confirmed in Figure 5 in Appendix D. Since $\psi_{\text{EN}}, \psi_{\text{DE}}$ are single linear layers (MLPs), we can bound the number of parameters by $pd + d^2 + 3d + dk$, with $k$ the number of label classes, in the (DD)-variant while in the (D)-variant we have $pd + 3d + dk$. Further ablation studies appear in Figure 4 in the Appendix showing that (DD) outperforms *sum* and GCN – especially in the lower hidden dimension regime – on real-world benchmarks with varying homophily.

**Synthetic experiments and ablation studies.**
To investigate our claims in a controlled environment we use the synthetic Cora dataset of [51, Appendix G]. Graphs are generated for target levels of homophily via preferential attachment – see Appendix D.3 for details. Figure 2 confirms the spectral analysis and offers a better understanding in terms of performance and smoothness of the predictions. Each curve – except GCN – represents one version of $\mathbf{W}$ as in 'methodology' and we implement eq. (11) with $\beta = 0$, $\mathbf{\Omega} = \mathbf{0}$. Figure 2 (top) reports the test accuracy vs true label homophily. *Neg-prod* is better than *prod* on low-homophily and viceversa on high-homophily. This confirms Proposition 3.1 where we have shown that the gradient flow can lead to a HFD dynamics – that are generally desirable with low-homophily – through the negative eigenvalues of $\mathbf{W}$. Conversely, the *prod* configuration (where we have an attraction-only dynamics) struggles in low-

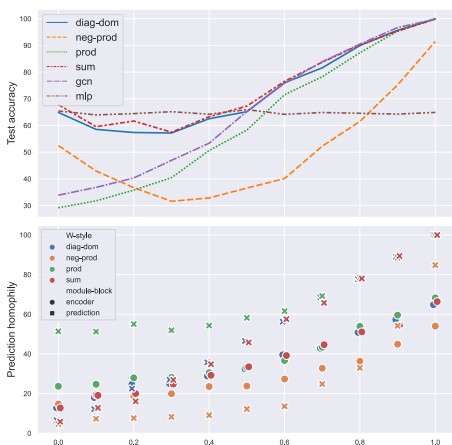

Figure 2: Experiments on synthetic datasets with controlled homophily.

homophily scenarios *even though a residual connection is present*. Both *prod* and *neg-prod* are 'extreme' choices and serve the purpose of highlighting that by turning off one side of the spectrum this could be the more damaging depending on the underlying homophily. In general though 'neutral' variants like *sum* and (DD) are indeed more flexible and better performing. In fact, (DD) outperforms GCN especially in low-homophily scenarios, confirming Theorem 4.3 where we have shown that without a residual connection convolutional models are LFD – and hence more sensitive to underlying homophily – irrespectively of the spectrum of $\mathbf{W}$. This is further confirmed in Figure 3.

In Figure 2 (bottom) we compute the homophily of the prediction (cross) for a given method and we compare with the homophily (circle) of the prediction read from the encoding (i.e. *graph-agnostic*). The homophily here is a proxy to assess whether the evolution is *smoothing*, the goal being explaining the smoothness of the prediction via the spectrum of $\mathbf{W}$ as per our theoretical analysis. For *neg-prod* the homophily after the evolution is lower than that of the encoding, supporting the analysis that negative eigenvalues of $\mathbf{W}$ enhance high-frequencies. The opposite behaviour occurs in the case of *prod* and explains that in the low-homophily regime *prod* is under-performant due to the prediction

| | Texas | Wisconsin | Cornell | Film | Squirrel | Chameleon | Citeseer | Pubmed | Cora |
|---|---|---|---|---|---|---|---|---|---|
| Hom level | **0.11** | **0.21** | **0.30** | **0.22** | **0.22** | **0.23** | **0.74** | **0.80** | **0.81** |
| #Nodes | 183 | 251 | 183 | 7,600 | 5,201 | 2,277 | 3,327 | 18,717 | 2,708 |
| #Edges | 295 | 466 | 280 | 26,752 | 198,493 | 31,421 | 4,676 | 44,327 | 5,278 |
| #Classes | 5 | 5 | 5 | 5 | 5 | 5 | 7 | 3 | 6 |
| GGCN | $84.86 \pm 4.55$ | $86.86 \pm 3.29$ | $85.68 \pm 6.63$ | $37.54 \pm 1.56$ | $55.17 \pm 1.58$ | $71.14 \pm 1.84$ | $77.14 \pm 1.45$ | $89.15 \pm 0.37$ | $87.95 \pm 1.05$ |
| GPRGNN | $78.38 \pm 4.36$ | $82.94 \pm 4.21$ | $80.27 \pm 8.11$ | $34.63 \pm 1.22$ | $31.61 \pm 1.24$ | $46.58 \pm 1.71$ | $77.13 \pm 1.67$ | $87.54 \pm 0.38$ | $87.95 \pm 1.18$ |
| H2GCN | $84.86 \pm 7.23$ | $87.65 \pm 4.98$ | $82.70 \pm 5.28$ | $35.70 \pm 1.00$ | $36.48 \pm 1.86$ | $60.11 \pm 2.15$ | $77.11 \pm 1.57$ | $89.49 \pm 0.38$ | $87.87 \pm 1.20$ |
| GCNII | $77.57 \pm 3.83$ | $80.39 \pm 3.40$ | $77.86 \pm 3.79$ | $37.44 \pm 1.30$ | $38.47 \pm 1.58$ | $63.86 \pm 3.04$ | $77.33 \pm 1.48$ | $90.15 \pm 0.43$ | $88.37 \pm 1.25$ |
| Geom-GCN | $66.76 \pm 2.72$ | $64.51 \pm 3.66$ | $60.54 \pm 3.67$ | $31.59 \pm 1.15$ | $38.15 \pm 0.92$ | $60.00 \pm 2.81$ | $78.02 \pm 1.15$ | $89.95 \pm 0.47$ | $85.35 \pm 1.57$ |
| PairNorm | $60.27 \pm 4.34$ | $48.43 \pm 6.14$ | $58.92 \pm 3.15$ | $27.40 \pm 1.24$ | $50.44 \pm 2.04$ | $62.74 \pm 2.82$ | $73.59 \pm 1.47$ | $87.53 \pm 0.44$ | $85.79 \pm 1.01$ |
| GraphSAGE | $82.43 \pm 6.14$ | $81.18 \pm 5.56$ | $75.95 \pm 5.01$ | $34.23 \pm 0.99$ | $41.61 \pm 0.74$ | $58.73 \pm 1.68$ | $76.04 \pm 1.30$ | $88.45 \pm 0.50$ | $86.90 \pm 1.04$ |
| GCN | $55.14 \pm 5.16$ | $51.76 \pm 3.06$ | $60.54 \pm 5.30$ | $27.32 \pm 1.10$ | $53.43 \pm 2.01$ | $64.82 \pm 2.24$ | $76.50 \pm 1.36$ | $88.42 \pm 0.50$ | $86.98 \pm 1.27$ |
| GAT | $52.16 \pm 6.63$ | $49.41 \pm 4.09$ | $61.89 \pm 5.05$ | $27.44 \pm 0.89$ | $40.72 \pm 1.55$ | $60.26 \pm 2.50$ | $76.55 \pm 1.23$ | $87.30 \pm 1.10$ | $86.33 \pm 0.48$ |
| MLP | $80.81 \pm 4.75$ | $85.29 \pm 3.31$ | $81.89 \pm 6.40$ | $36.53 \pm 0.70$ | $28.77 \pm 1.56$ | $46.21 \pm 2.99$ | $74.02 \pm 1.90$ | $75.69 \pm 2.00$ | $87.16 \pm 0.37$ |
| CGNN | $71.35 \pm 4.05$ | $74.31 \pm 7.26$ | $66.22 \pm 7.69$ | $35.95 \pm 0.86$ | $29.24 \pm 1.09$ | $46.89 \pm 1.66$ | $76.91 \pm 1.81$ | $87.70 \pm 0.49$ | $87.10 \pm 1.35$ |
| GRAND | $75.68 \pm 7.25$ | $79.41 \pm 3.64$ | $82.16 \pm 7.09$ | $35.62 \pm 1.01$ | $40.05 \pm 1.50$ | $54.67 \pm 2.54$ | $76.46 \pm 1.77$ | $89.02 \pm 0.51$ | $87.36 \pm 0.96$ |
| Sheaf (max) | $85.95 \pm 5.51$ | $89.41 \pm 4.74$ | $84.86 \pm 4.71$ | $37.81 \pm 1.15$ | $56.34 \pm 1.32$ | $68.04 \pm 1.58$ | $76.70 \pm 1.57$ | $89.49 \pm 0.40$ | $86.90 \pm 1.13$ |
| GRAFF (DD) | $88.38 \pm 4.53$ | $87.45 \pm 2.94$ | $83.24 \pm 6.49$ | $36.09 \pm 0.81$ | $54.52 \pm 1.37$ | $71.08 \pm 1.75$ | $76.92 \pm 1.70$ | $88.95 \pm 0.52$ | $87.61 \pm 0.97$ |
| GRAFF (D) | $88.11 \pm 5.57$ | $88.83 \pm 3.29$ | $84.05 \pm 6.10$ | $37.11 \pm 1.08$ | $47.36 \pm 1.89$ | $66.78 \pm 1.28$ | $77.30 \pm 1.85$ | $90.04 \pm 0.41$ | $88.01 \pm 1.03$ |
| GRAFF-timedep (DD) | $87.03 \pm 4.49$ | $87.06 \pm 4.04$ | $82.16 \pm 7.07$ | $35.93 \pm 1.23$ | $53.97 \pm 1.45$ | $69.56 \pm 1.20$ | $76.59 \pm 1.53$ | $88.26 \pm 0.41$ | $87.38 \pm 1.05$ |

Table 1: Results on heterophilic and homophilic datasets

being smoother than the true homophily. (DD) and *sum* variants adapt better to the true homophily. We note how the encoding compensates when the dynamics can only either attract or repulse (i.e. the spectrum of $\mathbf{W}$ has a sign) by decreasing or increasing the initial homophily respectively.

**Real world experiments.** We test GRAFF against a range of datasets with varying homophily [37, 33, 31] (see Appendix D.4 for additional details). We use results provided in [45, Table 1], which includes standard baselines as GCN [27], GraphSAGE [23], GAT [42], PairNorm [48] and recent models tailored towards the heterophilic setting (GGCN [45], Geom-GCN [31], H2GCN [51] and GPRGNN [13]). For Sheaf [5], a recent top-performer on heterophilic datasets, we took the best performing variant (out of six provided) for each dataset. We also include continuous baselines CGNN [44] and GRAND [10] to provide empirical evidence for Proposition 4.1. Splits taken from [31] are used in all the comparisons. The GRAFF model discussed in 'methodology' is a very simple architecture with shared parameters across layers and run-time smaller than GCN and more recent models like GGCN designed for heterophilic graphs (see Figure 5 in the Appendix). Nevertheless, it achieves competitive results on all datasets, performing on par or better than more complex recent models. Moreover, comparison with the 'time-dependent' (DD) variant confirms that by sharing weights across layers we do not lose performance. We note that on heterophilic graphs short integration time is usually needed due to the topology being harmful and the negative eigenvalues of $\mathbf{W}$ leading to exponential behaviour (see Appendix D).

## 6 Conclusions

In this work, we developed a framework for GNNs where the evolution can be interpreted as minimizing a multi-particle learnable energy. This translates into studying the interaction between the spectrum of the graph and the spectrum of the 'channel-mixing' leading to a better understanding of when and why the induced dynamics is low (high) frequency dominated. From a theoretical perspective, we refined existing asymptotic analysis of GNNs to account for the role of the spectrum of the channel-mixing as well. From a practical perspective, our framework allows for 'educated' choices resulting in a simple convolutional model that achieves competitive performance on homophilic and heterophilic benchmarks while being faster than GCN. Our results refute the folklore of graph convolutional models being too simple for heterophilic benchmarks.

**Limitations and future works.** We limited our attention to a *constant* bilinear form $\mathbf{W}$, which might be excessively rigid. It is possible to derive non-constant alternatives that are *aware* of the features or the position in the graph. The main challenge amounts to matching the requirement for local 'heterogeneity' with efficiency: we reserve this question for future work. Our analysis is also a first step into studying the interaction of the graph and 'channel-mixing' spectra; we did not explore other dynamics that are neither LFD nor HFD as per our definitions. The energy formulation points to new models more 'physics' inspired; this will be explored in future work.

**Societal impact.** Our work sheds light on the actual dynamics of GNNs and could hence improve their understanding, which is crucial for assessing their impact on large-scale applications. We also show that instances of our framework achieve competitive performance on heterophilic data despite being faster than GCN, providing evidence for efficient methods with reduced footprint.

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
