# OpenReview forum: "Graph Neural Networks as Gradient Flows"
_NeurIPS.cc/2022/Conference — NeurIPS 2022 Submitted_

### Official Review · Reviewer_PPbd · 2022-07-10

**Rating:** 6
**Confidence:** 4
**Soundness:** 4 excellent
**Presentation:** 3 good
**Contribution:** 3 good

**Summary:**

This paper introduces a new family of models, GRAFF, on graphs wherein graph features are transformed according to a dynamical system given by the negative gradient of an *energy functional*, which is parameterized and learned.  This amounts to a re-parameterization to focus on an energy function describing a discretized iterative update, instead of parameterizing the iterative update itself, as the most widely used GNN architectures do. This relation is properly studied in Section 4, where they show that GRAFF still includes many prior GNN model (up to the perhaps critically important matter of the non-linearity). However, this re-parameterization appears to offer more than just a reinterpretation of existing models. The primary value added explored in this work is to the analysis (and empirics) of the ability to handle heterophilious graphs.

**Questions:**

- What component(s) of GRAFF explain the inference speedup vis-à-vis GCN? Is it due to parameter sharing? Also, no details are given as to how this comparison was decided. Right now I cannot be sure that the comparison is apples to apples; maybe the GCN is a really massive model and the GRAFF is much smaller. More details on this would be great.
- Why just compare inference time? What about comparison of training time? It seems remiss not to include this, especially since the main paper simply mentions “run-time smaller than GCN” (line 359).
- The connection to spectral GNNs is interesting. Perhaps this suggests a path to developing expressive power results.

**Limitations:**

Yes.

**Strengths And Weaknesses:**

First, a congratulations to the authors on a nice piece of work that offers some new perspectives, and promising new directions. I enjoyed reading your work and certainly felt that I learned something in the process.

Below I discuss some of the things I especially liked in this work, as well as some of the concerns I have about certain aspects. Overall, I think this work contains some great conceptual components, but leaves open so quite important questions, particularly revolving around expressive power. The empirical evaluation is also relatively weak, and leaves me uncertain whether gradient flow models would enjoy widespread adoption. I will explain why I came to each of these beliefs in more detail below.

---

**Strong aspects:**

The community is moving towards a number of candidate approaches for circumnavigating the weaknesses of message passing networks. Although many ideas have already been given, the debate remains open. The idea of models following gradient flows is creative, and immediately prompted an “aha!” feeling. The community is in need of creative ideas like this, as you never know which will end up having a decisive impact.

On a technical level, it was pleasing to see the amenability of gradient flows to analysis of smoothing properties. The result in Section 3 on the Dirichlet energy functional were particularly interesting. It is quite unfortunate, however unfortunately typical, that the analysis doesn’t extent to non-linear activations (line 201).

---

**Weak aspects:**

A major missing piece of the picture is an understanding of the expressive power of the proposed gradient flow models. I found it particularly perplexing that line 202 mentions that no non-linearity is used in experiments. This raises a number of questions: are these models then of comparable expressive power to linear GNNs? Given that having no non-linearity doesn’t hurt performance does this just suggest that the empirical benchmarks considered just aren’t that challenging?

The paper claims several times that GRAFF models are “explainable”. The basis for this is that the model predictions can be understood by probing property of the energy functional. While this may turn out to be a useful point, the paper does not properly substantiate the claim. Indeed, there are no examples of any such “explanation” in practice. I would ask the author to either drop the “explainable” claim entirely (which isn’t critical in any-case, despite it’s prominent position in the explanation of ”why a gradient flow?”) or to clearly substantiate it, probably via an example.

Experimental evaluation is fairly limited. Table 1 is the main seat of comparisons to other models on node level classification tasks for varying levels of homophily. As billed, the strengths of GRAFF seem to emerge primarily in low homophily (high heterphily) graphs. However, the heterophilic graphs are very small: half only have a few hundred nodes, and the biggest graph considered—“Films”- has 7,600 nodes, and an MLP is a fairly competitive baseline on Films. All this means that the possible benefits to empirical methodology in the immediate future from seem unclear.

---

To conclude, although the empirics leave a number of question marks over the immediate empirical viability, the idea of graph models via gradient flows along an energy functional is elegant and thought provoking for me. The idea itself, plus the good exploration of the connection to smoothing, is enough to put me on the side of acceptance, but the limitations mentioned keep it only marginally so.


---

**Miscellaneous:**

- Clarity in certain places could be improved. For instance, Propositions 2.4 and 3.1 give explicit rates for the energy functional. Since (unless I am missing something!) the key point in both cases is that the models are high-frequency dominant, the rate itself seems to be more of an intermediate step towards this final HFD conclusion. Maybe it is a matter of personal taste but I would have hidden the gory details I the appendix.
- More generally, the paper is pretty notation heavy

---

> ### Author Response · Authors · 2022-07-30
> **Detailed response: part 1**
>
> We thank the reviewer for the very detailed feedback – which has inspired some further theoretical investigation on our side as reported in the general response and below – and for liking our paper. We encourage the reviewer to first consider reading the general response above where we touch many crucial points and we also list all the revisions already made to the main file and the SM. Below we address each specific point in detail, reporting $\textbf{your comments in quotations}$.
>
>  "It is quite unfortunate, however unfortunately typical, that the analysis doesn’t extent to non-linear activations (line 201)."
>
> To address this point, we have extended some partial analysis to the non-linear case and have added Proposition 3.2(line 199) and Lemma E.2 (line 978) to further clarify how even with pointwise activations we can maintain the duality between attraction and repulsion induced by the spectrum of the channel-mixing matrix $\mathbf{W}$ interpreted as a bilinear potential. This deserves further (but more involved) discussion that for the time being is beyond the purpose of this submission but $\textbf{we note that this is novel}$, since differently from previous works we are not simply considering the classical Dirichlet energy with respect to ReLU but instead a more general energy (wrt a more general class of activations) that can also magnify the high-frequency components.
>
> "This raises a number of questions: are these models then of comparable expressive power to linear GNNs?"
>
> We agree that this is an interesting point. So a few comments:
> - In some way, one may argue that the LFD/HFD characterizations introduced in our submission are already a measure for expressive power and that our linear framework minimizing a quadratic energy is able to generate both and so it is expressive in that sense.
> - We have addressed the expressive power concern in the general response but we reiterate the response here as well. In terms of linear gradient flow framework, note that we are residual meaning that the resulting polynomial has any term $\bar{\mathbf{A}}^{k}$ for $0 \leq k \leq m$, with $m$ number of layers (as explained in the new line 239 and the equation(27) line 792 in the SM) and indeed, our framework is not equivalent to collapsing the layers into a single one as SGCN given that we are residual. From a continuous point of view, the differential equation is linear but not the solution (which in fact will be exponential). We have also extended our theory to include pointwise non-linear activations that are guaranteed to still make the energy decrease along the solution; we hope that this addresses your concerns.
> - We also note that the source term may also lead to resonance effects and the various works [41,9,36] all explore – one way or another – this resonance phenomenon leading to better expressive power. We highlight how the energy approach can handle source terms as well so all the benefits apply to our setting too.
>
> "Given that having no non-linearity doesn’t hurt performance does this just suggest that the empirical benchmarks considered just aren’t that challenging?"
>
> The point about benchmarks has been thoroughly addressed in the general response that we invite you to consider. A further point we would like to emphasize here. The benchmarks are the same and only ones used by baselines (see general response) that are extremely more sophisticated/involved (and much slower) than ours and specifically designed to get to those same numbers as ours. It seems a bit unfair that we `have to pay a price' for showing that a residual linear model that is a gradient flow can match those performances given that the main purpose of our work is understanding GNNs and the channel-mixing role from a new light rather than proposing a new architecture. In any case, we are working on further checking the role of pointwise non-linear activation that preserves energy monotonicity as argued above and we’ll come back with results later.

---

> > ### Author Response · Authors · 2022-07-30
> > **Detailed response: part 2**
> >
> > "The paper claims several times that GRAFF models are “explainable”. The basis for this is that the model predictions can be understood by probing property of the energy functional. While this may turn out to be a useful point, the paper does not properly substantiate the claim. Indeed, there are no examples of any such “explanation” in practice. I would ask the author to either drop the “explainable” claim entirely (which isn’t critical in any-case, despite it’s prominent position in the explanation of ”why a gradient flow?”) or to clearly substantiate it, probably via an example."
> >
> > We agree that the term "explainable" (almost as in any case) is a bit ambiguous and in the revised version has been removed when redundant/unnecessary. We think that part of it though has been substantiated in the synthetic experiments where we could test how controlling specifically the spectrum of the channel-mixing affects the smoothness (homophily) of the prediction precisely as indicated by our theory. In this regard, we feel this is the `example’ you may be alluding here. More generally, explainable here is mostly relative to existing GNNs for which it is much harder to investigate a posteriori what is happening. For example, in a gradient flow framework one could analyze the spectrum of the learned channel-mixing and have an idea of whether the dynamics is going to be mostly smoothing (LFD) or sharpening (HFD), $\textbf{can something similar be said so easily otherwise}$?  Explicitly, say we approximate the spectral radius of the graph Laplacian by 3/2 (if it is larger, then it is even easier): if the most negative eigenvalue of the learned W is larger than twice the most positive one, then we are certain (mathematically) that the dynamics is going to be sharpening and concentrate more on the high frequencies. Viceversa, if the most positive eigenvalue is larger than the most negative, then we are certain we are going to have a smoothing dynamics (LFD).
> >
> > "Experimental evaluation is fairly limited. "
> >
> > We have thoroughly  addressed this in the general response.
> >
> > "However, the heterophilic graphs are very small: half only have a few hundred nodes, and the biggest graph considered—“Films”- has 7,600 nodes, and an MLP is a fairly competitive baseline on Films. All this means that the possible benefits to empirical methodology in the immediate future from seem unclear."
> >
> > This is not entirely accurate. Even with homophily, we are extremely close to the "top performant one" (if there is actually one in a meaningful way). Also, about immediate future applications, this holds for almost any GNN paper that focuses on node classification and even more so for the much more involved (and slower) baselines we compared with that were specifically designed to target heterophily. We think again this is not a fair criticism to us given all papers that have been published (see the list in the rebuttal). It cannot be a criticism to a faster and `simpler’ model to be competitive with more complicated ones where the latter use $\textbf{the very same benchmarks}$. Again, our paper though is mostly focussing on understanding theoretically modules like channel-mixing from this multi-particle point of view, introducing formulations as LFD and HFD to discuss expressive power and corroborate this with ablation.
> >
> > "the key point in both cases is that the models are high-frequency dominant, the rate itself seems to be more of an intermediate step towards this final HFD conclusion. Maybe it is a matter of personal taste but I would have hidden the gory details I the appendix"
> >
> > Somehow the point is about HFD but the convergence rate is important too. Something that usually is lost in standard references about asymptotic analysis is what the convergece rate (i.e. the second fastest term) is. In our analysis we provide explicit characterizations for that showing how it depends on the gaps of the spectra of the Laplacian and of the channel-mixing. This could potentially lead to better designs: for example, we want to slow down the HFD behaviour, then we need closer eigenvalues in the W-spectrum, and viceversa if we want to speed that up. To simplify a bit the discussion we have removed the explicit formula for $\epsilon_{\mathrm{HFD}}$ and reported in the SM, see also general revision comments above.

---

> > > ### Author Response · Authors · 2022-07-30
> > > **Detailed response: final part and conclusions**
> > >
> > > "What component(s) of GRAFF explain the inference speedup vis-à-vis GCN? Is it due to parameter sharing? Also, no details are given as to how this comparison was decided. Right now I cannot be sure that the comparison is apples to apples; maybe the GCN is a really massive model and the GRAFF is much smaller. More details on this would be great.
> > > Why just compare inference time? What about comparison of training time? It seems remiss not to include this, especially since the main paper simply mentions “run-time smaller than GCN” (line 359)."
> > >
> > > We answer both points here. The speed up is $\textbf{provable}$ and mainly due to the fact that the initial projections from higher dimensional raw features to smaller hidden dimension is done node-wise rather than edge-wise (this was argued in our complexity paragraph on line 306). Weight-sharing helps but it is secondary to the first point highlighted here. Concerning experiments on Figure 5 in SM we compare GCN and GRAFF with same hidden dimension (that is the x-axis of Figure 5) so we believe it is a $\textbf{fair comparison}$ by definition.
> > >
> > >
> > > We hope we have addressed all your concerns --  especially regarding $\textbf{theoretical analysis with activations}$ and the comparison with the many baselines and recent papers addressing $\textbf{the same problem}$ $\textbf{on the same}$ $\textbf{benchmarks}$ -- and we would appreciate if you raise the score; otherwise let us know of any other doubt/question and we are happy to address them in the discussion period.

---

> > > > ### Comment · Reviewer_PPbd · 2022-08-08
> > > > **Response to rebuttal**
> > > >
> > > > A big thank you for the detailed rebuttal both to myself, and other reviewers, which I have carefully read. I appreciate the efforts to clarify your contribution, which I continue to think is significant.
> > > >
> > > > Let me clarify and summarize my thinking on a couple of points:
> > > >
> > > > -  I am glad to see the new theoretical results for non-linear activations.
> > > >
> > > > - You are not 'paying a price' in terms of my accept/reject assessment for showing that linear activations suffice. Actually I think this observation makes a useful contribution towards demonstrating that the datasets considered by yourself and prior work are too simple. Reiterating my earlier comment, the fact that an MLP is general within a few percent of the best model is another angle on this point. So overall, _I don't hold the choice of datasets against you_ in any way, but do think that the area as a whole should move to more challenging benchmarks. There do exist such datasets (e.g., https://arxiv.org/pdf/2104.01404.pdf) which I hope you might consider for future projects.
> > > >
> > > > - I do not think that the synthetic experiment amounts to an adequate demonstration of your model being "explainable". Talking in the abstract, the phrase "explainable" has no precise meaning and is best avoided in a work that is otherwise rigorous in terminology, arguments etc. I am glad to hear you have removed some mentions to explainability, but notice that the word "explainable" or "explainability" still appears six times in the document. I strongly urge you completely remove all mention. I am not asking this for my benefit but yours, since the mention of explainability devalues the scientific seriousness of your work. I am not going to penalize you on this point since it is not related to the contribution of your paper, but I do feel strongly nonetheless.
> > > >
> > > > In all I remain in favor of this work being accepted, and will make this case during reviewer discussions. However I do not plan to raise my score further as I still think that the expressive power of these models is not clear, and the empirical results remain not especially strong.
> > > >
> > > > Best

---

> > > > > ### Author Response · Authors · 2022-08-09
> > > > > **Thanks for response and final comments.**
> > > > >
> > > > > Thank you for the response and finding our contribution significant. Some final points:
> > > > >
> > > > > - The analysis of non-linear activations in Proposition 3.2 and the whole Section E in the SM is quite novel in the GNN literature and in fact as you acknowledged it is much more common to have theoretical analysis restricted to the linear case. This effectively puts our framework into the expressive power landscape of other MPNNs; in general though, we are quite unsure about what you mean by "expressive power". If you look at all the references we shared above that are aimed at targeting heterophily, they do not have expressive power analysis (either at all or at least in a conventional way). In fact, our LFD and HFD characterizations arguably represent meaningful ways (but of course not exhaustive yet) of studying expressive power for node-classification tasks where one has also node-wise features rather than graph isomorphism tests.
> > > > >
> > > > > - Thank you for the reference, this will definitely be accounted for in future evaluations since we believe in this new way of thinking about GNNs as functionals and their minimal action. In this submission we were very much interested in the theory and in the simplest formalization of quadratic energy, but much more can be said and further and better evaluation is on our table when using more `sophisticated' gradient flows.
> > > > >
> > > > > - We agree on the ambiguity of the word explainable and accordingly have removed all its occurrences except one in the intro. This will be removed in a camera ready version where we will incorporate all the feedback to better rephrase the goals and contributions of our submission in light of the feedback we have received.
> > > > >
> > > > > - Concerning the empirical evaluation not being strong we again agree with you but need to point out that this is on par with several recent papers that do not share the level of theoretical investigation of our work. We are the $\textbf{first}$ proposing this new angle for studying GNNs, investigating the role of the channel-mixing spectrum and also (thanks to the feedback) proposing energy dissipation arguments when using non-linear activations.
> > > > >
> > > > > Once again, thank you for your time and for the suggestions to improve the paper.

---

### Official Review · Reviewer_c28L · 2022-07-11

**Rating:** 6
**Confidence:** 4
**Ethics Flag:** Yes
**Soundness:** 3 good
**Presentation:** 4 excellent
**Contribution:** 3 good

**Summary:**

In this paper, the evolution of the GNN is explained as learning attractive and repulsive forces in feature space by the positive and negative eigenvalues of a symmetric 'channel-mixing' matrix. According to the spectral analysis of the solutions, gradient flow graph convolutional models result in a dynamic dominated by graph high frequencies, which is desirable for heterophilic datasets. Moreover, the authors present structural constraints on common GNN architectures, allowing them to be interpreted as gradient flows. We perform extensive ablation studies to verify our theoretical analysis and demonstrate the comparative performance of simple and lightweight models on real-world homophilic and heterophilic datasets.

**Questions:**

If two particle (two nodes) is repulsive to each other, will both feature blow up (going to infinity) as the time of the dynamic system increases? How can you ensure all the feature on every node is bounded with the system is evolving?




**Ethics Review Area:**

["I don’t know"]

**Limitations:**

The authors discussed the limitation of their works and their social impact.

**Strengths And Weaknesses:**

Strengths:

In this paper, the author gives a new perspective on GNN in terms of the particle system, which explain why the original GNN does not work well on heterophilic datasets and also analysis Dirichlet energy change in the dynamic system.

The whole paper's structure is clear and easy to follow. Several adequate experiments are used to verify the author's statement.

Weaknesses:

When the graph size is increasing, is this GNN also computation feasibly? As this model needs to compute the eigendecomposition of the graph Laplacian, when the graph size is increasing, it should be hard to compute.

---

> ### Author Response · Authors · 2022-07-30
> **Detailed response**
>
> We thank the reviewer for their feedback and for believing that in our paper we "give a new perspective on GNN in terms of the particle system, which explain why the original GNN does not work well on heterophilic datasets and also analysis Dirichlet energy change in the dynamic system" and for thinking that "the whole paper's structure is clear and easy to follow and that several adequate experiments are used to verify the author's statement".
>
> We kindly ask the reviewer to check the general response before where we address several important points. Below, we further reply to each specific comment raised in this review. We report your $\textbf{feedback/comments in quotations}$.
>
> "As this model needs to compute the eigendecomposition of the graph Laplacian, when the graph size is increasing, it should be hard to compute."
>
> There is a $\textbf{misunderstanding}$ here that we hope to clarify. We $\textbf{never}$ have to $\textbf{compute}$ the $\textbf{eigendecomposition of the graph Laplacian}$ or not even an approximate one and we are not sure which line of the paper was misleading about this point. The $\textbf{eigenvectors are used}$ from a $\textbf{theoretical level only}$ to derive results. The framework acts as a $\textbf{classical message passing}$ using the sparsity of the input graph and indeed $\textbf{it is as fast as classical GCN}$ (as reported in Figure 5, line 950 in the SM) and much faster than spectral methods.
>
> "If two particle (two nodes) is repulsive to each other, will both feature blow up (going to infinity) as the time of the dynamic system increases? How can you ensure all the feature on every node is bounded with the system is evolving?"
>
> This is an interesting question. The linear dynamics is in principle unbounded – albeit there are non-linear ways to prevent solution to blow-up (see for example the form of the non-linear gradient flow in Section B.3 of the SM). However, the fact that the dynamical system is unbounded is not intrinsically bad for a variety of reasons:
> - the problem of unbounded dynamics only emerges as one approaches the limit of infinite layers (i.e. infinite integration time) and of course the system is always well-defined for each finite time
> - On a heterophilic graph, separating adjacent particles that are `opposite’ to each other $\textbf{fast}$ can be beneficial and this could be achieved thanks to the negative eigenvalues of the channel mixing which induce repulsion.
> - Even simple models like (S)GCN can be unbounded. In fact, the norm of the solution can blow up (again in the infinite time limit) as soon as some eigenvalue of the channel mixing has absolute value larger than one. We also invite you to read the new paragraph in line 278 added in our revised submission.
>  - Note also that the pre-image of the one-hot encoding of the softmax in fact is by design never bounded.
>
> Since the $\textbf{only weakness was based on the wrong premise}$ that we hope we have addressed (once again we never compute eigendecomposition of graph Laplacian, our model is a sparse MPNN like GCN), we'd appreciate if you raise your score -- otherwise let us know any other specific concerns and we're happy to address them in the discussion period.

---

### Official Review · Reviewer_4YFR · 2022-07-11

**Rating:** 4
**Confidence:** 3
**Soundness:** 2 fair
**Presentation:** 3 good
**Contribution:** 2 fair

**Summary:**

In this work, authors present GRAFF, a gradient flow based graph neural network in which the evolution of GNN is represented as minimizing the combination of attractive and repulsive interactions of a multi-particle system. Detailed theoretical characterization in terms of a parametric Dirichlet energy, a general parametric energy and spectral analysis is performed.

**Questions:**

It has been shown by earlier works that energy formulation results in a loss landscape that has a large number of local minima and hence gradient based minimization results in a poor solution. In contrast, deep neural networks with large number of parameters have flat minima and the loss landscape is connected by level set. Reading these together, it is unclear whether the energy-based formulation with simpler formulation loses the advantages of the landscape that a deep-learning architecture has. Could this also be the reason why the performance of the GRAFF is not superior in comparison to other SOTA models? Authors should investigate this.

**Limitations:**

The experiments has primarily focussed on one aspect while studying on several datasets with varying hetero/homophily. To evaluate the true performance of the approach, several other experiments on varying downstream tasks are required. in addition, a closer analysis on the loss landscape is required to understand the nature of the minima and saddle points.

**Strengths And Weaknesses:**

The work is well-written and clearly presented. It builds on similar ideas as outlined in several previous works such as, for instance, [27]. While the presentation in the work is good, the idea in itself is fairly intuitive and simple and has been discussed in contexts of neural networks and physics (related works, see: Landscape and training regimes in deep learning, M. Geiger; L. Petrini; M. Wyart, Physics Reports. 2021-04-16. Vol. 924, p. 1-18. DOI : 10.1016/j.physrep.2021.04.001.). Further, the empirical experiments reveal that the results are comparable with the existing approaches, but not necessarily better.

---

> ### Author Response · Authors · 2022-07-30
> **Detailed response**
>
> We thank the reviewer for their feedback and for finding that the work is "clearly written and well presented". We encourage the reviewer to first see our general response above. Below we address each specific point raised in the review more in detail.
>
> $\textbf{Your comments are reported in quotations}$.
>
> "While the presentation in the work is good, the idea in itself is fairly intuitive and simple and has been discussed in contexts of neural networks and physics (related works, see: Landscape and training regimes in deep learning)"
>
> While we agree that the idea of gradient flows is not new and in fact dates back by centuries, we have not seen these approaches applied in the context of Graph Neural Networks. In particular, although the reference is very interesting and has already been added in our revised version, we think that there might be some $\textbf{misunderstanding}$ here that we hope to clarify. We are not applying energy arguments to understanding the loss – our discussion is $\textbf{not about the loss}$. The energy is a functional of the node representations that will determine how the nodes get updated in a GNN. We do not see any strong comparison with the reference here and most importantly we do not see why "the energy-based formulation with simpler formulation loses the advantages of the landscape that a deep-learning architecture has". We $\textbf{cannot lose any advantage}$ given by $\textbf{deep-learning}$ architectures since $\textbf{our framework is a deep learning}$ one: this is a point that seems to have been misunderstood. We are not changing the loss and indeed our framework has the same advantages as other deep learning frameworks on graphs like GCN in terms of loss landscape.
>
>
> To further emphasize the (possible) point of confusion: the reviewer said in addition "a closer analysis on the loss landscape is required to understand the nature of the minima and saddle points". Our energy formulation is not about the loss, it is the energy that regulates the update equations (i.e. the forward) we are not modifying the backward pass. Concerning the experiments on homophily/heterophily, this is a standard practice that has been used exactly as for our work by [47], [4], [5], [42], [11], [15], [28], [37], [45]. We invite you to see our general point about benchmarks in the general response section.
>
>
> Since the $\textbf{main (only) weakness raised seemed to be based on a wrong premise}$ that hopefully we have managed to clarify how it does not apply to our work, we'd appreciate if you raise your score -- otherwise let us know any other specific concerns and we're happy to address them in the discussion period.

---

> > ### Comment · Reviewer_4YFR · 2022-08-07
> > **Thank you for your response.**
> >
> > Thank you for clarifying the points in detail. I agree with the authors' comment partially. However, I believe there has been some communication gaps as detailed below.
> > > The energy is a functional of the node representations that will determine how the nodes get updated in a GNN.
> >
> > 1.  In the formulation presented by the authors, the node update is carried out based on the gradient of a parametric energy functional. Specifically, the authors show that the node features can be updated along the direction of energy minimization $\dot{\varepsilon}_\theta(\textbf{F}(t))=-|| \nabla_\textbf{F} \varepsilon_\theta(\textbf{F}(t))||^2$). My specific question was regarding the nature of the landscape of the **learned energy** as a function of the parameters $\theta$ and $F$, which was inadvertently written as loss landscape.
> > 2. Additionally, the node features are not unique and is simply a representation of the node. Hence, multiple node features can result in the same output label even in a well-trained GNN. This corresponds to degenerate states in an energy landscape, that is, states having the same energy but different configurations (read node features in this case). But these regions can have different local curvature and hence the nature of the gradient depends highly on the nature of the energy landscape.
> >
> > In short, I think exploring the nature of the learned energy functional in terms of its landscape, minima, and saddle points, is crucial to develop an understanding of the limitations and usefulness of the approach. I hope I have been able to clarify my questions to the authors. I am not expecting any additional experiments. However, it'll be helpful if the authors can respond to these queries.
> >
> > > Concerning the experiments on homophily/heterophily...
> > I understand the focus is on node classification. Thank you for the detailed explanation.

---

> > > ### Author Response · Authors · 2022-08-07
> > > **About energy landscape**
> > >
> > > Thanks for clarifying this further and for taking the time to explain this to us. We address your points about the minimization process and how the learnable parameters have an impact on what we can learn (along with features). Note that below $\mathbf{W}$ is learnable and hence represents parameters. In summary, our theoretical analysis gives us a very good picture of the energy landscape since we can both determine how the learnable parameters $\mathbf{W}$ affect where our normalized solution converges to and how fast such convergence is happening (as a function of both graph information and learnable $\mathbf{W}$).
> > >
> > > In fact, we would like to emphasize that to the best of our knowledge this is the $\textbf{first work}$ in GNN highlighting convergence of the solution to states that are not in the Laplacian kernel and can hence carry information from input features and graph structure (indeed we can explain how the learnable parameters $\mathbf{W}$ can guide such convergence through its spectrum).
> > >
> > >  More details to follow:
> > >
> > > - Our theoretical analysis is precisely about the impact of the learnable channel-mixing $\mathbf{W}$ on the minimization process and indeed we have quite a good understanding of how such landscape looks like. Proposition 3.1 shows that if the learned parameters have more mass on the negative eigenvalues (in the precise way stated in the Proposition), then the normalised solution will converge to the span of the largest frequency eigenvector of the Laplacian which hence represents (if we let the dynamics run for infinite time until convergence) the landscape where my minimum lives. Where exactly in this subspace the solution converges to is a function of the initialized features. The latter are partly given by the problem and partly learned via an encoding. Conversely, if we have more mass on the positive eigenvalues of $\mathbf{W}$, then the normalized solution will converge to the span of the lowest frequency eigenvector of the Laplacian. So in general one has a pretty clear picture of what the dynamics is accomplishing and where the minima are going to sit.
> > >
> > > - An important point though is that while for losses one would like to converge to lower (ideally unique) minima, in the case where node features are updated following a gradient flow this may not be the case. This is similar to the ODE case where the existence of a Lyapunov function tells you something about fixed points but you may want to learn (or tune) the integration time and perhaps even halt the evolution before convergence. To put it more concretely: in some cases (homophilic graphs), $\textit{some}$ smoothing is beneficial (so you would most likely see that the learned channel-mixing $\mathbf{W}$ has more positive eigenvalues to mostly induce attraction). However if you run it for too long and hence arrive to convergence to the minima (after normalization), then you will sit in the Laplacian kernel meaning that the only information left to separate node representations are degrees (which is the classical over-smoothing problem). This of course is not desirable, which is also why architectures like GCN are generally shallow (short integration time).
> > >
> > > - Another subtle but important point is the initial condition. In principle one has a node-wise initial encoding step that can learn where in the energy profile we should start. This can be very useful. To give you an example, assume that the largest frequency Laplacian eigenvector $\mathbf{v}$ is not that good for classification, but the second highest frequency one $\mathbf{u}$ is. Then the initial encoding could learn to give us an initial condition that is orthogonal to $\mathbf{v}$ such that if we learn $\mathbf{W}$ with more negative eigenvalues as before, then the normalized solution converges to a minima that is in the span of $\mathbf{u}$. In principle this would also help us into controlling where in the span of $\mathbf{u}$ we are converging to i.e. as you raised in your second point which degenerate state we end up converging to. Note that this can also be aid by the decoding step that ideally would learn to choose among the degenerate states the one with better separation power.
> > >
> > >
> > > We thank the reviewer for bringing this point up which we will investigate even further in the future. We hope that at least regarding the scope of our submission, we have clarified the point about the landscape of the minimization process.

---

### Official Review · Reviewer_13DY · 2022-07-11

**Rating:** 4
**Confidence:** 5
**Soundness:** 3 good
**Presentation:** 2 fair
**Contribution:** 2 fair

**Summary:**

The paper suggests a new graph neural network architecture that can be seen as a gradient descent minimization of a learnable energy function. The authors use channel-mixing matrices with mixed eigenvalues to infuse high frequencies into the architecture dynamics. Many existing architectures fall into this framework. Three variants of the new architecture are presented.

**Questions:**

- The gradient of Eq (4) looks like Grad^TW^TWGrad(F). Why is Eq (5) the way it is? Please clarify in the text.

- I am confused by the derivation from (6) to (7). First, if the rest of the paper uses the energies in (7), why introduce the energies in (4)-(6)? How do (6) and (7) relate? This is not clear. More importantly, is it the same W in both equations? It does not seem so. This is very confusing. Please consider revising.

- Line 202-203: Are the authors saying that there is no role for the non-linearities in graph neural networks? But that is the most important aspect of a neural network’s definition (otherwise, the whole network collapses to a single linear operator). It does not make sense. Maybe try other experiments? CNNs, for example, sure require non-linearities for image classification. Maybe test this hypothesis on graph classification? Maybe shape classification (ModelNet40)?

- Line 209: by linearized GNNs, do the authors mean with identity activation? Because there are no activations in the following equations. But - this is not the traditional use of the word "linearized", which is traditionally used for a Taylor approximation. Please revise.

- Line 228 – why do the authors introduce tilde{W}, and then set it as identity? Can’t tilde{W} be chosen better? And if so - why introduce this matrix?

- Line 230 – Why are Omega and W shared across the layers? Traditionally, at least in CNNs, one learns a variety of channel-mixing convolution operators. What is different here? This negates the common practice of neural networks and requires explanation and evaluation.

Line 235 – the authors have T, tau, and the number of layers. How do we choose T or tau? Is it a hyperparameter? Do the authors choose tau to be small enough to ensure the stability of the Euler scheme?

Line 236 – The authors state that they can include the non-linearities in the encoder and decoder layers only. Will that be the equivalent to having a non-linearity in Eq. (12)? I do not understand why. Again, that is against the fundamental aspect of neural networks – the use of non-linear activations to express complex functions.

Lines 244-245: Symmetry being a key requirement. When other works learn the channel-mixing operators, they do not enforce symmetry. So – this is important only for looking at architectures as a gradient flow. Further, in lines 247-248: indeed (13) can be seen as a generalization of the mentioned methods (with identity activation), but none of these learn symmetric matrices. Also, as the authors note, GAT does not have a symmetric attention matrix. So, is symmetry really important? Will we get better networks if we enforce symmetry in the learning?

In continuation to the previous point: the symmetrization in lines 304. In my opinion, this whole concept of symmetry or not should be evaluated in extensive experiments, but I do not see such experiments in the paper.

Lines 282-284: Essentially, when choosing negative eigenvalues for W, you indeed reverse the time integration. But then, the Euler method that is used is known to become unstable. Isn’t that a problem with the whole approach?

Lines 290: The authors state that linear GNNs achieve competitive performance on real-world data sets. Again, having linear layers negates the whole concept of NNs. This cannot be a conclusion that holds for all data sets and tasks. Given the rather limited scope of experiments, I would say that this is a too strong statement here.

Lines 301-302: The authors choose W to be diagonal and random and do not train over it. Why is that? Won’t we get better results if we train over W? This means that there are no non-linearities in the network, and there are no learnable channel-mixing parameters. So, except the encoding and decoding layers, it’s essentially a classical algorithm. How do the authors explain that?

Lines 301: the random W. How does the choice of a random W influence the results? Are the achieved accuracies stable? Or do the authors see large deviations between different runs?

Line 375: This is yet another strong sentence, given the rather limited experimental study in the paper.


**Limitations:**

yes.

**Strengths And Weaknesses:**

Strengths
1) The paper is rich in theoretical insights.
2) Unlike previous works, this is the first work that analyzes the channel mixing matrix.

Weaknesses:
1) The experiments are rather limited. The authors show only node classification, where it is customary to show more experiments. The work of GCNII, for example, shows PPI, and two cases of node classification (semi and fully supervised).  Also, an example on a large dataset (e.g., OGBN-Arxiv) is also important. Most importantly, it is not clear how well the method works on graph classification tasks (e.g., the TUD data sets), without the non-linearities in the layers. Given the strong claims made by the authors regarding those non-linearities (see below), showing these experiments is essential in my opinion.

2) I find only a single data set where GRAFF yields the best performance. Overall, this method does not improve the SOTA.

3) There is very little discussion about over-smoothing in this work (only in the paragraph in line 249). At the end, the authors do not show if their method is over-smoothing or not. Ideally, the authors would provide accuracies for a variety of number of layers and show that the accuracy does not degrade (e.g., see GCNII).

4) There are questionable choices for the architecture (e.g., no non-linearities) that are accompanied by too strong statements without backing them up. See details in the questions section. In particular, the authors do not show that indeed adding and removing the non-linearity has no influence on the accuracy.

5) The writing of the paper is hard to follow. I would say that the presentation (i.e., notation and language) can be simplified to make this paper more reader-friendly.

6) I could not find how many layers were used in Table 1.

7) Missing citation from the previous Neurips:
Zhou, K., Huang, X., Zha, D., Chen, R., Li, L., Choi, S.-H., and Hu, X. Dirichlet energy constrained learning for deep graph neural networks. Advances in Neural Information Processing Systems, 34, 2021.
This paper also discusses the Dirichlet energy throughout the layers.

8) Missing citation from ICLR 2022:
How attentive are graph attention networks? (GATv2)
The conclusion of this paper is that having a more non-linear (in some sense) attention matrix improves the accuracy and training stability over GAT. How does the conclusion in this work align with the findings in GATv2? Note that there are additional data sets in GATv2, which may be more challenging and require non-linearities in the layers.

---

> ### Author Response · Authors · 2022-07-30
> **Detailed response part 1**
>
> We thank the reviewer for their feedback and for finding our paper "rich in theoretical insights", and the first to "analyze the channel mixing matrix". We kindly ask to first check our $\textbf{general response}$. Below we address each point raised separately and more in detail. $\textbf{We report your comment in quotes first}$: note that partly as a consequence of that the response will be relatively long.
>
> "The experiments are rather limited. The authors show only node classification, where it is customary to show more experiments. The work of GCNII, for example, shows PPI, and two cases of node classification (semi and fully supervised). Also, an example on a large dataset (e.g., OGBN-Arxiv) is also important. Most importantly, it is not clear how well the method works on graph classification tasks (e.g., the TUD data sets), without the non-linearities in the layers. Given the strong claims made by the authors regarding those non-linearities (see below), showing these experiments is essential in my opinion."
>
> The point about experiments and benchmarks has been addressed in the general response. We would like to reiterate that:
>
> - Works like GCNII or the suggested reference about constrained Dirichlet energy do $\textbf{not}$ investigate GNN performance on heterophilic datasets (the homophily of ogbn-arxiv is $\textbf{0.80}$)
> - Generally, works that are mostly interested in frequency response (smoothing vs sharpening effect) – as the ones listed in the general response – never test on large graphs and only focus on node-classification (homophily and frequency response are not very meaningful for graph-level tasks, nor are there such established specific benchmarks) with the inclusion of heterophilic baselines.
> - The experimental evaluation we report follows that used in most recent papers studying GNNs in heterophilic settings again see [47], [4], [5], [42], [11], [15], [28], [37], [45] as discussed in the general response.
> - Finally, our work – as acknowledged – is indeed mostly theoretical and provides an understanding of common elements of GNN models. Our extensive synthetic experiments and ablation studies fully support our theory.
>
> "I find only a single data set where GRAFF yields the best performance. Overall, this method does not improve the SOTA."
>
> We respectfully disagree with this comment and what this might entail. First, the performance of the top k models (usually k ~ 3) on almost all datasets is $\textbf{extremely close}$ and even defining what ‘SOTA’ means here is a gray area. Second, our paper describes a theoretical framework allowing to design more interpretable GNN architectures (in the sense that they minimize a well-understood energy) and make more educated architectural choices. In particular, we show both theoretically and experimentally that very simple architectures (linear residual GCN with shared symmetric layer parameters) can perform in heterophilic settings on par with much more complex SOTA models -- which is an interesting finding.
> We also emphasize an $\textbf{important point}$: in this work there is no separation between the theory section and the implementation, meaning that the model tested is precisely a discretized gradient flow (we have even removed intra-layer dropout to be as close as possible to the theoretical equations).
>
> "There is very little discussion about over-smoothing in this work (only in the paragraph in line 249). At the end, the authors do not show if their method is over-smoothing or not. Ideally, the authors would provide accuracies for a variety of number of layers and show that the accuracy does not degrade (e.g., see GCNII)."
>
> $\textbf{We find this comment a little worrying}$: line  249 (in the revised version now line 255) refers to an $\textbf{entire paragraph including the main theorem}$ of our paper which shows when and how the channel-mixing matrix has the power – thanks to its negative eigenvalues – to induce a high-frequency dominant dynamics which therefore avoid over-smoothing. To some extent, our whole work is about a better analysis of the smoothing and over-smoothing effects both in the finite time case (convergence rate) and asymptotic one. Note that we even have a formal definition of over-smoothing in line 118. In fact, our theoretical analysis in Theorem 4.3 offers a further theoretical justification in terms of the spectrum of the channel-mixing matrix for why methods like GCNII that introduce a residual connection avoid over-smoothing. We kindly ask the Reviewer to read again the paragraph starting at line 255 along with the new paragraph at line 278 and tell us if there are any doubts about our smoothing/sharpening analysis.

---

> > ### Author Response · Authors · 2022-07-30
> > **Detailed response part 2**
> >
> >
> > "At the end, the authors do not show if their method is over-smoothing or not. Ideally, the authors would provide accuracies for a variety of number of layers and show that the accuracy does not degrade (e.g., see GCNII)"
> >
> > This is a very important point we would like to clarify, which is also one of the key messages of our paper. We believe that there has been some confusion in the literature describing over-smoothing as a degradation of performance. Clarifying this somewhat vaguely used concept is a reason why we proposed a formal definition for it (line 117; see also [7] and [31]). Note for example, that even a dynamics that by design magnifies high frequencies (i.e. the opposite of over-smoothing) like $\dot{\mathbf{F}}(t) = \boldsymbol{\Delta}\mathbf{F}(t)$ can lead to extreme performance degradation if the underlying graph is homophilic for example. To further emphasize why Theorem 4.3 is in fact a more accurate and general depiction of over-smoothing, we have added a new paragraph in line 278 which we invite you to read. Having said that:
> >
> > - Experimentally we use prediction homophily pre and post diffusion as a measure of oversmoothing. This is much more aligned with the actual problem of over-smoothing as in `smoothing too much’ so that the predicted homophily is (much) higher than the true one. We have fully confirmed in the ablation studies (line 314) that negative eigenvalues of the channel mixing matrix induce repulsion and indeed lead to generally low homophily predictions i.e. they cannot oversmooth as proven rigorously in Proposition 3.1 (line 181) for the continuous case and Theorem 4.3 (line 260) for the discrete case.
> > - In general, heterophilic graphs are much more challenging in terms of going deeper and GCNII for example only ran on homophilic ones. Indeed, you can see from our Table 1 that GCNII generally seems to suffer on heterophilic graphs. The fact that you can go deeper does not necessarily help. On the other hand, we have also proved in Theorem 4.3 that over-smoothing (as LFD dynamics characterized in our paper) cannot be avoided if you remove residual connections which again is the reason why GCNII generally works better than GCN.
> >
> > "There are questionable choices for the architecture (e.g., no non-linearities) that are accompanied by too strong statements without backing them up. See details in the questions section. In particular, the authors do not show that indeed adding and removing the non-linearity has no influence on the accuracy"
> >
> >
> > This point has been addressed in the general response. We further emphasize here the following:
> > - We have proved in the new Proposition 3.2 (line 199) that one can have standard non-linear activations acting pointwise on top of our framework and still maintain the interpretation of energy decreasing along the solution. Further details in the new Section E of the SM.
> > - All our baselines are effectively non-linear (we use nonlinear encoder/decoder, and only the graph propagation is linear), so a substantial comparison with non-linear GNN models is in fact already provided.
> >
> > "The writing of the paper is hard to follow. I would say that the presentation (i.e., notation and language) can be simplified to make this paper more reader-friendly"
> >
> >
> > We thank the reviewer for the feedback, but would appreciate specific suggestions of what we might improve. We also notice other Reviewers were happy with our writing, and that we included a preliminary notation paragraph at the beginning of section 2 where we have introduced most conventions and notations used throughout the paper to help the reader. We have already simplified the notation/discussion in a few places as detailed in the general response about revised versions.
> >
> > "I could not find how many layers were used in Table 1."
> >
> > Section D5 in the SM has the hyperparameter choices for all the datasets, including the integration time and step size inferring number of layers used.
> >
> > "Dirichlet energy constrained learning for deep graph neural networks. Missing citation."
> >
> > We thank the reviewer for pointing us to this interesting reference that had already been added. A few minor comments:
> >
> > - The claim that Dirichlet energy going to infinity may lead to over-separation is not accurate and that can be explained thanks to the new characterization of LFD dynamics we have introduced. Please consider the new paragraph in line 278 explaining why our Theorem 4.3 provides a better characterization of smoothing dynamics than simply looking at Dirichlet energy (rather than using our LFD characterization).
> > - In connection to our general point about benchmarks, we would like to emphasize that the given reference tested $\textbf{only}$ node classification $\textbf{tasks on homophilic datasets}$ further supporting the point that the benchmarks considered “challenging” are either large (homophilic) graphs or (smaller) heterophilic graphs.

---

> > > ### Author Response · Authors · 2022-07-30
> > > **Detailed reponse: part 3**
> > >
> > > "Missing citation from ICLR 2022: How attentive are graph attention networks? (GATv2) The conclusion of this paper is that having a more non-linear (in some sense) attention matrix improves the accuracy and training stability over GAT. How does the conclusion in this work align with the findings in GATv2? Note that there are additional data sets in GATv2, which may be more challenging and require non-linearities in the layers"
> > >
> > > We thank the reviewer for pointing the citation to us that has been added. Concerning the role of non-linear maps, please refer to our general point above and the new theoretical paragraph in line 194. About GATV2: their findings apply to the attention mechanism and in particular to the collapsing into a single layer of the query projection and head attention one. This is not equivalent to what we do since in our framework (as explained in the new line 239 and the equation(27) line 792 in the SM). We refer though to how we can include non-linear activations too by maintaining the interpretation as discussed in the general response. Note that we are not studying an attention framework and we are not stating that non-linearities are not needed.
> > >
> > > $\textbf{Questions}$
> > >
> > > "The gradient of Eq (4) looks like Grad^TW^TWGrad(F). Why is Eq (5) the way it is? Please clarify in the text."
> > >
> > > This is proved in the SM, see line 717-719 and the review about Kronecker product in Section A.2.
> > >
> > > "I am confused by the derivation from (6) to (7). First, if the rest of the paper uses the energies in (7), why introduce the energies in (4)-(6)? How do (6) and (7) relate? This is not clear. More importantly, is it the same W in both equations? It does not seem so. This is very confusing. Please consider revising."
> > >
> > > The introduction of the energy in (4)-(6) is pedagogical since we show how one could easily generalize approaches that were used with success in both geometry and image processing to graphs. However, we also show in Proposition 2.4 that such framework would always lead to a smoothing process (and over-smoothing in the asymptotic limit if the kernel of H is zero) that may be not desirable to deal with heterophily. Accordingly, we define a more general energy in equation (7) where we have now two matrices Omega and W not necessarily equal. This is a more general energy and does not have to be equal to the previous one. Based on your feedback, we have rephrased this part more explicitly in the revised version, let us know if this reads better now (lines 161--163).
> > >
> > > "Line 202-203: Are the authors saying that there is no role for the non-linearities in graph neural networks? But that is the most important aspect of a neural network’s definition (otherwise, the whole network collapses to a single linear operator). It does not make sense. Maybe try other experiments? CNNs, for example, sure require non-linearities for image classification. Maybe test this hypothesis on graph classification? Maybe shape classification (ModelNet40)?"
> > >
> > > No, we simply state that we did not find large improvements in our framework by making it non-linear given that this always has a price in terms of speed and interpretation. In any case, inspired by this point, we have added a reply in the general response. We have now a new theoretical justification for using activation functions like (tanh, ReLU, arctan..) (see Proposition 3.2 line 199) in our framework without losing the interpretation given by the channel mixing matrix W in terms of attraction and repulsion. We are running further tests and ablations. Note again that differently from SGCN that instead removes non-linear maps and collapses the GNN to a single layer, it is not true that our framework is equivalent to a single linear layer since we have a residual connection (as explained in the new line 239 and the equation(27) line 792 in the SM).
> > >
> > > "Line 209: by linearized GNNs, do the authors mean with identity activation? Because there are no activations in the following equations. But - this is not the traditional use of the word "linearized", which is traditionally used for a Taylor approximation. Please revise."
> > >
> > > Yes that is correct. We again emphasize though that the differential equation is linear (i.e. we have a residual connection) but the solution won’t be (indeed we are approximating a matrix exponential solution in the discrete case).
> > >
> > > "Line 228 – why do the authors introduce tilde{W}, and then set it as identity? Can’t tilde{W} be chosen better? And if so - why introduce this matrix?"
> > >
> > > The reason for introducing \tilde{W} is to show that this can be done in the gradient flow framework i.e. the source term induced by \tilde{W} can be derived from a source term at the energy level. When evaluating the framework, we wanted to keep the number of parameters low so we decided to choose \tilde{W} = I. In principle, other choices could be better, however our paper is $\textbf{not about fine tuning a specific model}$ to get a marginal improvement over SOTA.

---

> > > > ### Author Response · Authors · 2022-07-30
> > > > **Detailed response: part 4**
> > > >
> > > > "Line 230 – Why are Omega and W shared across the layers? Traditionally, at least in CNNs, one learns a variety of channel-mixing convolution operators. What is different here? This negates the common practice of neural networks and requires explanation and evaluation."
> > > >
> > > > The sharing across layers is due to the fact that these symmetric matrices represent the potentials in the energy and accordingly are taken to be time-independent, otherwise one would not be able to conclude that the energy is being minimized along the GNN. We will emphasize this point better, but this is again a choice due to the physics model by which we are inspired. The end goal is using this framework to investigate the role and importance of the channel-mixing matrices. This is checked in Table 1 line 347 where the last row GRAFF -timedep(DD) is a variant of our framework where we do not share weights across layers. As you can see this is not better and in fact, sometimes marginally worse despite the larger number of parameters.
> > > >
> > > > "The authors have T, tau, and the number of layers. How do we choose T or tau? Is it a hyperparameter? Do the authors choose tau to be small enough to ensure the stability of the Euler scheme?"
> > > >
> > > > When discretizing the differential equations using explicit (forward Euler) scheme with fixed step size tau and integration time T, the ratio L=T/tau corresponds to the number of layers (see l. 235-236) and it is a hyperparameter as for standard GNNs. We refer the Reviewer to Section D5 in the SM where we have reported all the hyperparameters, including integration time T and step size tau for all the nine datasets.
> > > >
> > > > "The authors state that they can include the non-linearities in the encoder and decoder layers only. Will that be the equivalent to having a non-linearity in Eq. (12)? I do not understand why. Again, that is against the fundamental aspect of neural networks – the use of non-linear activations to express complex functions."
> > > >
> > > > The non-linearities can be used in either the encoding map or the decoding one or both. This is not equivalent to having a non-linearity in equation (11), as typically done in GCN, and we did not state that. We show an architecture implementing a nonlinear map from raw node features to node labels, where the only nonlinearity is in node-wise encoder/decoder, whereas the graph propagation part is linear (a discretized differential equation, without layer-wise nonlinearity). Our experiments suggest such an architecture can perform very well (on par with much more complex SOTA architectures, including heterophilic datasets). Please see our general reply about non-linearity too and especially the point where we discuss how – based on your feedback and the Reviewer $\textcolor{red}{PPbd} – we have introduced a new theoretical justification for using pointwise non-linearities (Proposition 3.1 in line 199) in this energy framework.
> > > >
> > > > "Symmetry being a key requirement. When other works learn the channel-mixing operators, they do not enforce symmetry. So – this is important only for looking at architectures as a gradient flow. Further, in lines 247-248: indeed (13) can be seen as a generalization of the mentioned methods (with identity activation), but none of these learn symmetric matrices. Also, as the authors note, GAT does not have a symmetric attention matrix. So, is symmetry really important? Will we get better networks if we enforce symmetry in the learning?"
> > > >
> > > > As the Reviewer correctly notes, symmetry is a constraint under which the architecture is a gradient flow minimizing a well-understood energy, making it more interpretable and allow for more educated architectural choices. Our paper is mostly focused on providing an understanding of the channel-mixing in terms of attraction and repulsion, however note that one could possibly leverage our insight to better initialize a given channel-mixing (in terms of eigenvalues for example) which was beyond the scope of our paper. We did not show that enforcing symmetry always leads to better networks. GAT does not have a symmetric matrix and in fact GAT cannot be interpreted as a gradient flow: if you think about physics and dynamical systems in general, not all differential equations are gradient flow of an energy, but many important ones, especially the ones coming from physics, are. Most importantly, GAT is one of our baseline and is consistently beaten by a large margin when the graph is heterophilic. This shows that the removal of the symmetry constraint – even if more general in principle – is not necessarily helpful. Furthermore, we refer the Reviewer to ablation studies in the SM that thoroughly investigate what we gain/lose by enforcing symmetry vs standard (non-linear) GCN (see Figure 3 in the SM).

---

> > > > > ### Author Response · Authors · 2022-07-30
> > > > > **Detailed response: part 5 and important conclusion**
> > > > >
> > > > > "The authors state that linear GNNs achieve competitive performance on real-world data sets. Again, having linear layers negates the whole concept of NNs. This cannot be a conclusion that holds for all data sets and tasks. Given the rather limited scope of experiments, I would say that this is a too strong statement here"
> > > > >
> > > > > Please see our general response about non-linearities. We did not state this to be the case on all real world datasets but on 9 datasets (with different levels of homophily) that the community has been using for a while now (see all the baselines). Also, very popular papers like SGCN [40] have already argued that removing non-linear activations in a GNN is not always bad.
> > > > >
> > > > > "The authors choose W to be diagonal and random and do not train over it. Why is that? Won’t we get better results if we train over W? This means that there are no non-linearities in the network, and there are no learnable channel-mixing parameters. So, except the encoding and decoding layers, it’s essentially a classical algorithm. How do the authors explain that?"
> > > > >
> > > > > This is just one of the baselines, the simplest one that satisfies our framework and is indeed not better performant than the one where we train the channel-mixing over the majority of datasets. We are not claiming that this is the best one and in fact did not observe/report to be the best one. It was more a conceptual experiment to test the lightest possible instance of our framework where we simply ask the encoding and decoding to learn to re-align the features based on graph and random diagonal attraction/repulsion uniformly sampled. We find it surprising that this is enough on the smaller heterophilic graphs.
> > > > >
> > > > > "How does the choice of a random W influence the results? Are the achieved accuracies stable? Or do the authors see large deviations between different runs?"
> > > > >
> > > > > We did not observe large deviations, but again we emphasize that this is just a conceptual experiment. We are not claiming that avoiding to learn the channel-mixing is always helpful, which is why we have reported experiments with the dense, learnable, diagonally dominant channel-mixing configuration that perform better.
> > > > >
> > > > > "Line 375. This is yet another strong sentence, given the rather limited experimental study in the paper."
> > > > >
> > > > > Our experiments are in line with most recent papers studying heterophily. The complex benchmarks we used are the heterophilic graphs that recent papers specifically targeting this issue have been considering, see [47], [4], [5], [42], [11], [15], [28], [37], [45] and general response.
> > > > >
> > > > >
> > > > > $\textbf{Important conclusion}$
> > > > >
> > > > > We hope that we have addressed all points and that the reviewer will let us know of any outstanding doubt. We kindly ask to re-consider the score based on both the general response and our replies given that
> > > > > - most of the questions were already addressed in the SM
> > > > > - this is a theoretical work that has still shown competitive performance w.r.t much more complicated architectures despite empirical evaluation not being its focus
> > > > > - that we believe the questions about non-linear maps have been addressed in the context of the energy framework from a theoretical standpoint in the general response (see the new Section E in the SM and the new Proposition 3.2 in line 199)
> > > > > - and that we have provided ample evidence for why papers in this community interested in the heterophily setting and frequency response have used the very same benchmarks we adopted which cannot be a genuine reason for criticism in order to enforce fairness.

---

> > > > > > ### Comment · Reviewer_13DY · 2022-08-08
> > > > > > **Response to authors**
> > > > > >
> > > > > > Dear Authors,
> > > > > >
> > > > > > Thank you for the detailed and long response. I still find it hard to recommend the acceptance of your paper.
> > > > > >
> > > > > > My point 1 remains. I agree that other papers showed only these experiments. But - they did not have such strong claims regarding the non-linearities being unnecessary. Non-linearities are not necessary for node classification, which you show, but probably not for other problems. Again - this is against the whole concept of deep neural networks.  The authors compare their method to GRAND and GCNII which show more results. Actually, GRAND shows completely different results than in this paper, and the results in this paper are new runs, as far as I can tell. Why don't the authors compare their method to GRAND (Table 1 or 2) and GCNII (Tab 2) on the splits reported there? Overall - to make such strong claims, the authors need to show more empirical evidence, in this reviewer's opinion. See also experiments in GraphCon and GATv2, that the authors can choose from.
> > > > > > Also, for some reason, when opening the paper GPRGNN, different and better results are presented: see table 2 here: https://arxiv.org/pdf/2006.07988.pdf (e.g., see Texas and Cornell). Is there an explanation for that?
> > > > > >
> > > > > > My point 2 also remains: I'm not sure that being top 3 on one task is in line with the empirical evaluation that is required in NeurIPS. I will let the other reviewers and AC decide on that.
> > > > > >
> > > > > > My point 3 also remains. As stated in the original review, I read the theorem in the original line 249. I also agree that one should not confuse performance degradation and over-smoothing. However, the authors do not show how their method behaves when adding more layers, as shown in many other works. Maybe the proof somehow misses something that we cannot find, or, maybe there is a difficulty in the learning process when adding more layers. Theorem 4.3 should be empirically validated, as done in other works. See Fig 2 in GraphCon, Tab 1 in EGNN, Tab 3 in GCNII, Tab 2 in GGCN, Fig 2 in GRAND, etc.
> > > > > >
> > > > > > Regarding the D variant: The D variant shows significantly better accuracy results, but the authors claim in their rebuttal that it is only a conceptual experiment. How can the best-performing method be a conceptual experiment only? The authors also do not address the variability of the D variant which is uninitialized between -1 to 1, and is not trained. As the authors discuss, the sign of the diagonal entries determines the type of interaction between nodes. Clearly, this has an impact on the action of the GNN, and therefore it is not convincing to me that various random initializations do not change the results.
> > > > > >
> > > > > > Regarding strong statements re non-linearities mentioned in point 1: The final statement, in conclusion, claims that complex datasets can be approached with simple networks. However, the tested datasets are rather small and simple, and additional experiments with complex datasets should be evaluated to reach such conclusions.
> > > > > >
> > > > > > I still think that the writing is hard to follow.
> > > > > >
> > > > > > Because of the remaining issues above, I keep my original score.
> > > > > >
> > > > > > Sincerely,
> > > > > > Reviewer 13DY

---

> > > > > > > ### Author Response · Authors · 2022-08-09
> > > > > > > **Thanks for response and final comments**
> > > > > > >
> > > > > > > Thank you for your further detailed response. A couple of final important points:
> > > > > > >
> > > > > > > - "But - they did not have such strong claims regarding the non-linearities being unnecessary. " If you look for claims in our paper, we never claimed that non-linear maps are never needed. We simply observed (as corroborated by our ablations) that on these datasets -- which are the same used by many other recent papers targeting heterophily -- non-linear activations may be removed without paying a price in performance. To further align with your feedback, we have revised the pdf and replaced the only occurrence of "complex benchmarks" with "heterophilic benchmarks". If you search for other mentions of non-linear maps not being useful or other strong claims, you won't find any in our main file. Most importantly, our new theoretical analysis also confirms that one can use non-linear activations and still retain the energy dissipation interpretation, which is generally non-trivial in dynamical systems. Therefore the new theory (Proposition 3.2) effectively
> > > > > > >  $\textbf{makes all the discussion}$ $\textbf{about linear vs non-linear redundant}$ since we can use non-linear maps in this framework -- we hope you can see that.
> > > > > > >
> > > > > > > - About the GPRGNN results, they use an easier split of 60/20/20 rather than the more commonly adopted 48/32/20 which papers like GGCN, Sheaf, and us rely on.
> > > > > > >
> > > > > > > - About being top 3: this is not an empirical work and conferences like NeurIPS are not just about beating baselines and we hope the reviewers agree on that given that arguably not all papers share the same level of theoretical analysis and investigation we provided. The purpose of this submission is highlighting a new way of thinking about GNNs where we parametrize an energy rather than an equation. As experimental analysis, we showed that equations that are precisely those given to us by the theory can be competitive on benchmarks commonly adopted in literature.
> > > > > > >
> > > > > > > - About the role of depth, most papers that stack many layers have time dependent weights that can suppress later contributions. This is expectable and indeed very often it is a case of maintaining the same performance with deeper architectures and not achieving much better numbers. In our theoretical analysis we actually do not expect that adding many layers will be beneficial because it's never the case that converging to either low-frequency or high-frequency eigenspaces of $\boldsymbol{\Delta}$ would be optimal for the given classification task. In fact, note that as argued in a different response, our analysis can be seen as studying of the energy landscape and highlights what the dynamics will converge to in infinite time (and how fast). We agree that general over-smoothing experiments are relevant and we plan to add those in the non-linear setting described in Section B.3 where one can go deeper in a principled manner while stile minimizing an energy with now a different landscape. This is a broader scope that we are reserving for future work.
> > > > > > >
> > > > > > > - The D variant is not generally better performant. It works a bit better on the small heterophilic datasets. Note that we have a learnable component in the encoding and decoding step so effectively given uniformly sampled eigenvalues from -1 to 1 we can always learn to re-order the features so tat the negative valued entries have repulsion and the positive ones have attraction so one would not expect much variation wrt the uniform sampling in such range (given the number of points sampled, usually 64). However, this is not enough on the larger heterophilic datasets where learning a dense channel-mixing is significantly better usually.
> > > > > > >
> > > > > > > - Regarding strong statements, we have addressed this point above.
> > > > > > >
> > > > > > > - Regarding the writing, we put a lot of efforts into conveying the theoretical message without diluting the mathematical details to avoid statements that are more accessible but also more vague and less transparent.
> > > > > > >
> > > > > > > Thank you for your time.

---

### Author Response · Authors · 2022-07-30
**General response part 1**

We thank the reviewers for finding that our paper is `rich in theoretical insights’, a ‘nice piece of work that offers some new perspectives, and promising new directions’, that is ‘well-written and clearly presented’, and that the work ‘ contains some great conceptual components’. We address here a few general but $\textbf{crucial}$ questions/doubts raised from the reviewers along with some $\textbf{key misunderstandings}$ that we hope to clarify. We hope that the reviewers revisit their scores in the light of our response.


$\textbf{Important disclaimer}$: We have revised the main file and SM based on the feedback and the new theoretical results including non-linear activations. All references below to equations and lines refer to the $\textit{revised version}$. Below you can also find a detailed and granular list with all modifications.

$\textbf{A few words on the goal}$:  The main goal of our paper consists in studying a new framework where GNNs minimize an energy with emphasis on its theoretical implications. This allows us to study the role of the channel-mixing and provide theoretical results in terms of smoothing vs sharpening dynamics induced by its spectrum, convergence rate and asymptotic behaviour. In fact, we emphasize how our results are more granular and more explicit than classical over-smoothing ones currently available in literature [26,27,7] and differently from those fully explain the role of the residual connection from the spectral perspective of the channel-mixing.


$\textbf{Benchmarks and empirical evaluation}$: Despite our discussion is general, the main underlying problem we are interested in is the frequency response of the GNN with associated performance $\textit{on heterophilic graphs}$, something that is becoming of increasing interest for the community. This is a list of references included in our paper that, for the great part, have the specific and unique goal of proposing models that work well with heterophily: [47], [4], [5], [42], [11], [15], [28], [37], [45]. $\textbf{They all have experiments on node classification task only}$ and using the $\textbf{very same datasets we have tested on}$ (some of the references on fewer datasets actually, while [45] has 2 extra homophilic datasets but 1 heterophilic dataset less). None of them test on $\textbf{graph-level tasks}$ for which $\textbf{the notions of homophily and frequency are less meaningful}$. Therefore we believe $\textbf{our experiments are aligned with recent papers and baselines}$ that are interested in the same problem as ours; moreover, with the exception of [5] and [15], none of the papers above arguably shares the same theoretical flavour and analysis as ours. We also note how although papers like [41], [8], [36] propose node-classification tasks on larger graphs – usually a single one –, the latter are $\textbf{homophilic}$ and indeed such references never test on heterophilic datasets. In fact, Table 1 in our paper compares with [8] and [41] for example, emphasizing how they are not suitable to handle heterophily. Accordingly, $\textbf{we believe our experiments to be extensive and sufficient}$ – especially considering that reviewers have acknowledged the main theoretical nature of our work.

$\textbf{The role of non-linear activations}$: Reviewers $\textcolor{blue}{13DY }$ and $\textcolor{red}{PPbd}$ have raised questions about the role of non-linear activations and associated evaluation. Some important preliminary remarks:
- We do not state that in the general graph learning landscape non-linear activations are not needed.
- The fact that in GNNs one can (sometimes) suppress non-linear activations without serious issues has been already observed, see for example the $\textit{highly popular}$ SGCN paper [43].
- Our framework is composed of a node-wise encoding block, a diffusion block, and a node-wise decoding block. In principle, both encoder and decoder can be chosen as non-linear MLPs, meaning that the $\textbf{overall architecture can indeed be nonlinear}$.
- The fact that the diffusion block (equation (11), line 234) is linear does $\textbf{not}$ mean that the GNN collapses to a single layer due to the residual connection term. More precisely, we refer to lines 790-795 of the $\textbf{revised SM}$ (the list of modifications is detailed below). Since we are discretizing a linear ODE, the solution is an approximation of an exponential map.

---

> ### Author Response · Authors · 2022-07-30
> **General response part 2**
>
> $\textbf{New theoretical results concerning non-linear activations}$: In response to Reviewer $\textcolor{red}{PPbd}$ question about expressive power and some comments by Reviewer $\textcolor{blue}{13DY}$, we included $\textbf{Proposition 3.2 line 199}$ and a new Section E in the SM investigating how a non-linear pointwise activation would fit this framework. In a nutshell: we prove that if we activate equations (11) (line 234) with a non-linear map $\sigma$ belonging to a large class of functions (including common choices like ReLU, $\arctan$, $\tanh$..), then the learnable energy $\mathcal{E}^{\mathrm{tot}}$ is still $\textbf{decreasing along the solution}$. This allows us to retain the interpretation of $\textbf{W}$ as inducing attraction and repulsion since the energy has not changed and is still decreasing. We have also added $\textbf{Lemma E.2 (line 978)}$ to check how in a simple diagonal case we maintain the same smoothing vs sharpening analysis. In principle then, we could have $\textbf{non-linear activations and keep the}$ same $\textbf{physics oriented approach}$ and interpretation where a learnable multi-particle energy is decreasing along the GNN. We believe this deserves further investigation and we reserve that for future work.
>
> $\textbf{Concerning the level of challenge of the benchmarks}$: We selected baselines in Table 1 specifically designed to perform well on heterophilic graphs (and in fact note the ones like GAT, GRAND, CGNN for example that are not and suffer significantly on heterophilic graphs). We show we are extremely competitive with much slower and more sophisticated baselines despite a simpler framework (however we again emphasize that we are $\textbf{not equivalent to a single linear layer}$ and that as $\textbf{per the new Proposition 3.1}$ we could also $\textbf{use non-linear activations}$). We restate the different baselines like GGCN, GPRGNN, H2GCN, Geom-GCN, Pair-Norm, Sheaf, all use the same task and same datasets for evaluation. A further important point concerning evaluation: in this work there is $\textbf{no separation between the theory}$ section and $\textbf{the implementation}$, meaning that the model tested is precisely a discretized gradient flow (we have even removed intra-layer dropout to be as close as possible to the theoretical equations).
>
> $\textbf{Some important misunderstandings}$. A few general points reviewers raised as weak points that are misunderstandings we would like to clarify:
> - Lack of discussion on over-smoothing $\textcolor{blue}{13DY}$: the whole paper is in some regard about the smoothing effect and how the channel-mixing is able to steer away diffusion from over-smoothing thanks to the negative eigenvalues as proved in the main Theorem 4.3. We have emphasized this point further in the revised version, as explained in the response below about modifications.
> - All points raised from $\textcolor{blue}{13DY}$ about choice of integration time, step sizes and other hyperparameters as well as derivation of equations are $\textbf{already addressed in the SM}$: please see detailed individual response.
> - It seems the review of $\textcolor{green}{4YFR}$ raised as only/main weakness what we believe is a misunderstanding concerning what our energy is; they asked what we lose "$\textit{compared to deep learning frameworks}$".  We clarify with reviewer $\textcolor{green}{4YFR}$ that ours $\textbf{is a deep learning framework}$ and the parametrised energy we use concerns the $\textbf{forward pass and not the backward}$ w.r.t to the loss optimisation. See also detailed response.
> - Reviewer $\textcolor{orange}{c28L}$ raised as a weakness that our framework may not scale to large graphs given we need to compute the graph Laplacian eigenvectors. This is $\textbf{not}$ the case. The SVD decomposition of the graph Laplacian $\textbf{is not required and the eigenvectors are only used in our theoretical analysis}$. In fact, our model is a sparse MPNN that is as fast as GCN and much faster than spectral methods (see Figure 5 in SM for a runtime comparison with GCN using same hidden dimension).

---

> > ### Author Response · Authors · 2022-07-30
> > **Revised main document and SM: detailed list of modifications**
> >
> > This is a detailed list of the modifications to the main file and SM (both now appear in their revised versions) based on the reviewers' feedback:
> >
> > - New references [50] and [8] (suggested by $\textcolor{blue}{13DY}$) and [6] (suggested by $\textcolor{green}{4YFR}$) have been added
> > - We have reformulated the introduction of $\mathcal{E}^{\mathrm{tot}}$ in lines (161--164) to address concerns about clairity raised from $\textcolor{blue}{13DY}$.
> > - Removed explicit formula for $\epsilon_{\mathrm{HFD}}$ and moved to the SM line 760 to reduce notations as suggested by $\textcolor{red}{PPbd}$
> > - The old paragraph about non-linear gradient flow has been moved to SM (line 780). Instead we have a $\textbf{new paragraph}$ in line 194 containing the $\textbf{new Proposition 3.2}$ about energy dissipation when using non-linear activation functions we considered based on feedback from both $\textcolor{blue}{13DY}$ and $\textcolor{red}{PPbd}$.
> > - We have a new Section E in the SM to discuss non-linear activation functions applied to a GNN Gradient Flow dynamical system.
> > - We have added two further bullet points in line 239 to explain why a linear discrete gradient flow is not equivalent to collapsing the MPNN into a single layer and that thanks to the new Proposition 3.2 we could also activate the equations with pointwise non-linear maps without losing most of the physics inspired interpretation.
> > - We have removed in a few instances the word `explainable' where not essential or ambiguous as suggested by $\textcolor{red}{PPbd}$ as for example in line 252.
> > - We have moved the previous paragraph about the edge sign flipping to the SM to have a less packed discussion about the implications of our main Theorem 4.3
> > - Added a new paragraph in line 278 commenting about the subtle but fundamental difference about over-smoothing and LFD in light of Theorem 4.3. Our contribution extends to unbounded channel-mixing spectral radius and shows that even though technically we are not over-smoothing we are still always LFD meaning that it is just a problem of global scale.

---

> > > ### Author Response · Authors · 2022-08-02
> > > **Final ablation studies about non-linear activations**
> > >
> > > We have included here a table with further ablation studies regarding the performance of GRAFF with and without non-linear activations, along with comparing it with GCN (and the different steps to go from GCN to GRAFF).
> > >
> > >
> > > Experiment details:
> > >
> > > | dataset | **GCN** | **1) +enc/dec** | **2) +residual** | **3) share weight** | **4) W symmetric** | **5) GRAFF linear** | **6) GRAFF DD linear** | **7) GRAFF DD non linear** |
> > > |---|---|---|---|---|---|---|---|---|
> > > | **Chameleon** | 61.93 ± 1.96 | 62.19 ± 2 | 66.14 ± 1.62 | 65.94 ± 1.9 | 66.34 ± 2.27 | 66.8 ± 2.28 | 69.34 ± 1.6 | 69.21 ± 1.13 |
> > > | **Citeseer** | 70.92 ± 2.46 | 70.95 ± 2.73 | 71.45 ± 2.11 | 71.05 ± 1.59 | 70.18 ± 2.07 | 70.5 ± 2.32 | 71.21 ± 2.44 | 71.1 ± 1.3 |
> > > | **Cora** | 81.63 ± 1.17 | 80.76 ± 1.67 | 81.07 ± 1.67 | 81.09 ± 1.85 | 80.73 ± 1.56 | 80.22 ± 1.76 | 81.62 ± 1.33 | 80.81 ± 1.85 |
> > > | **Squirrel** | 40.51 ± 1.33 | 43.01 ± 1.8 | 45.31 ± 1.83 | 46.3 ± 1.46 | 46.46 ± 1.92 | 47.07 ± 1.76 | 52.42 ± 1.81 | 54.59 ± 1.53 |
> > >
> > > We choose two homophilic real-world datasets Cora and Citeseer and two heterophilic datasets Chameleon and Squirrel, repeating the series of augmentations  1) add an encoder/decoder. 2) add a residual connection. 3) share the weights of $\mathbf{W}$ and $\boldsymbol{\Omega}$ across time/layers. 4) symmetrize $\mathbf{W}$ and $\boldsymbol{\Omega}$. 5) remove the non-linearity between layers, as described in the GCN ablation in SM Section D.3 to transition from GCN to GRAFF with a sum symmetric matrix.
> > >
> > > In the table we note column “4 - $\mathbf{W}$ symmetric” is equivalent to GRAFF non-linear with ReLU activation. In addition we try the version of GRAFF linear with diag-dom symmetric matrix (6) and a pointwise $\tanh$ nonlinearity (7).
> > >
> > > To give representative hyper-parameters for every data set we search over the space; lr {0.0001, 0.001, 0.005}, decay {0.0, 0.001, 0.005}, time {2, 3, 4}, step size {0.5, 1.0}, hidden dimension {64}. We take the best average performance of 10 splits, using the geom-gcn splits for the heterophilic datasets and random splits for the homophilic datasets consistent with the ablations in SM Section D.4.
> > >
> > > The new ablations provide a more explicit comparison between GRAFF (in its linear gradient flow formulation) and non-linear baselines like GCN and GRAFF activated with pointwise non-linear maps. As claimed, there is no (significant) performance deterioration between linear GRAFF and non-linear baselines (in particular its non-linear activated version). We again emphasize the following:
> > >
> > > - The suppression of non-linear activations in the context of GNNs has already been studied in the highly popular SGCN reference [43].
> > > - Linear GRAFF is not equivalent to a single linear layer due to the residual connection. Indeed, we have explicitly derived in the SM (lines 790--795) that linear GRAFF after m layers corresponds to an m-degree polynomial with all powers of the normalized adjacency entering the polynomial expansion.
> > > - The full model has encoding and decoding blocks that can be implemented using MLPs, hence making the $\textbf{whole map}$ $\textbf{raw-features} \rightarrow \textbf{labels}$ generally $\textbf{non-linear}$.
> > >
> > >
> > > A final important point in connection to the new theory we have added in the revised document (Proposition 3.2 line (199) and Section E in the SM): in principle, we have shown that one can activate the linear gradient flow using many common non-linear maps (ReLU, $\arctan$, $\tanh$) while preserving the physics inspired interpretation of the channel-mixing $\textbf{W}$ as a potential inducing attraction and repulsion along edges. Therefore, one can fully $\textbf{leverage the}$ $\textbf{expressive power}$ $\textbf{of non-linear maps in this multi-particle energy framework}$. The experiments here simply confirm that on the benchmarks commonly used by all the recent references we compared with in the context of heterophilic graphs (as amply argued in the previous general responses) there is no significant performance drop by removing the activation.
> > >
> > >
> > > Once again, we are happy to engage in discussion and clarify any standing doubt.

---

### Meta-Review · Area_Chair_w1Ns · 2022-08-29

**Recommendation:** Reject
**Confidence:** Less certain

**Metareview:**

The authors present a graph neural network for heterophilic data using gradient flows. The proposed architecture is quite simple...large sections of the architecture are fully linear dynamical systems rather than neural networks, and still achieve roughly SotA results on standard graph learning benchmarks. There was a significant amount of disagreement between the reviewers. Some seemed to think the strength of mostly linear methods meant that the benchmarks were too easy, but these are standard graph neural network benchmarks. A simple model performing well is not a negative, and can often be useful for puncturing hype (e.g. https://arxiv.org/abs/2206.13211). Simple architectures can also be useful for providing analytic insights which might get obscured in more complex models. Some reviewers seemed concerned about the scaling of certain tools (e.g. graph Laplacian eigenvectors), but these tools are only used for analysis, not for training. Nevertheless, I feel that there were enough general concerns around the paper that I have a difficult time recommending acceptance. Even if the purpose of the paper is primarily to drive analytic insights rather than achieve SotA results on big benchmarks, I would recommend the authors to show how these analytic insights can be used to improve models on big datasets to strengthen the paper.

**Award:**

No

---

### Decision · Program_Chairs · 2022-09-14

Reject